# AGENTIC REINFORCED POLICY OPTIMIZATION

**Guanting Dong**[1*]**, Hangyu Mao**[2*]**, Kai Ma**[3]**, Licheng Bao**[3]**, Yifei Chen**[1]**, Zhongyuan Wang**[3]
**Zhongxia Chen**[3]**, Jiazhen Du**[3]**, Huiyang Wang**[3]**, Fuzheng Zhang**[3]**, Guorui Zhou**[3†]
**Yutao Zhu**[1]**, Ji-Rong Wen**[1]**, Zhicheng Dou**[1†]
[1]Renmin University of China, [2]Institute of Microelectronics, Chinese Academy of Sciences
[3]Kuaishou Technology
{dongguanting, dou}@ruc.edu.cn, maohangyu@ime.ac.cn

## ABSTRACT

Large-scale reinforcement learning with verifiable rewards (RLVR) has proven effective in harnessing the potential of large language models (LLMs) for single-turn reasoning tasks. In realistic reasoning scenarios, LLMs often rely on external tools to assist in task-solving processes. However, current RL algorithms typically employ trajectory-level rollout sampling, consistently neglecting the fine-grained exploration of multi-turn tool-call steps. To bridge this gap, we propose **A**gentic **R**einforced **P**olicy **O**ptimization (**ARPO**), a novel agentic RL algorithm tailored for training multi-turn LLM-based agents. Our preliminary experiments reveal that LLMs frequently exhibit increased uncertainty after tool-call steps, as evidenced by higher entropy in the distribution of generated tokens. Motivated by this, ARPO incorporates an entropy-based adaptive rollout mechanism, encouraging the policy model to adaptively branch sampling during high-entropy tool-call rounds, thereby promoting step-level exploration of latent tool-use behaviors. By integrating an advantage attribution estimation, ARPO enables LLMs to internalize advantage differences in stepwise tool-use interactions. Experiments across 13 challenging benchmarks demonstrate ARPO's superiority over trajectory-level RL algorithms. Remarkably, ARPO achieves improved performance using only half of the tool-use budget required by existing methods, offering a scalable solution for aligning LLM-based agents with real-time dynamic environments. Our codes are released at https://github.com/RUC-NLPIR/ARPO.

## 1 INTRODUCTION

Recently, large-scale Reinforcement Learning with Verifiable Rewards (RLVR) has demonstrated strong potential in unleashing the capabilities of frontier large language models (LLMs), showcasing impressive performance across various single-turn reasoning tasks (OpenAI, 2024; DeepSeek-AI et al., 2025; Team et al., 2025b; Qwen et al., 2024; Yang et al., 2025). However, in open-ended reasoning scenarios (Putta et al., 2024; Shridhar et al., 2020; Qin et al., 2024), LLMs should not only cultivate long-horizon planning and adaptive decision-making skills, but also engage in dynamic, multi-turn interactions with external tool environments. To address these challenges, **Agentic R**einforcement **L**earning (**Agentic RL**) (Singh et al., 2025b; Zhang et al., 2025; Team et al., 2025a) enables LLMs to autonomously interact with external tool environments during RL training, shifting the training paradigm from static task solving to the landscape of dynamic agent-environment interactions (Silver et al., 2017; Wen et al., 2024; Qian et al., 2025a; Dong et al., 2026).

Current agentic RL algorithms typically perform trajectory-level exploration during the rollout phase (Shao et al., 2024; Yu et al., 2025), sampling complete tool-use trajectories with predefined special tokens and assigning rewards based solely on the final output. To further address tool overuse and sparse reward issues (Qian et al., 2025b), recent studies have proposed more refined reward functions to better align tool-use behavior (Wang et al., 2025a; Bai et al., 2025). Despite some

---

*Equal contribution.
†Corresponding author.

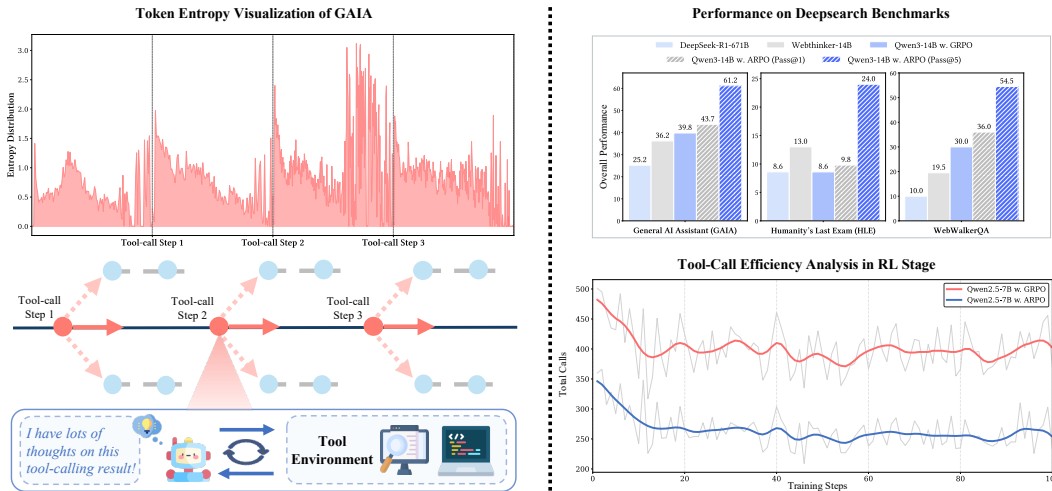

Figure 1: Overview of tool-use token entropy exploration and ARPO algorithm performance. **Left:** High entropy observed in the LLM following tool usage. **Right:** LLM performance comparison on deep search tasks using ***only 1k RL samples***, along with a comparison of training tool-use budgets.

progress, these optimizations often overlook a key aspect of training LLM-based agents: the multi-turn interaction loops between the LLM and the tool environment (Wang et al., 2025d; Xue et al., 2025; Jiang et al., 2025; Feng et al., 2025b). Unlike the single-turn reasoning paradigm, the multi-turn tool-use loop offers LLMs informative, real-time feedback in real-time. This highlights the necessity of discovering effective stepwise tool-use behaviors.

To gain an insight into such step-level behaviors, we draw inspiration from a series of entropy-based studies (Wang et al., 2025b;c; Zheng et al., 2025b) and quantify the token entropy distribution of LLM-based agents during deep search tasks. As illustrated in Figure 1 (left), the initial tokens generated after each round of tool-call feedback consistently exhibit high entropy. This indicates that external tool calls significantly increase uncertainty during LLM reasoning, revealing latent tool-use behaviors that remain underexplored (Ruan et al., 2023; Li et al., 2025h; Chen et al., 2025c). Unfortunately, current trajectory-level RL methods overemphasize complete rollout sampling comparisons, neglecting the exploration of fine-grained behavior in high-entropy tool-use steps (Xiong et al., 2024; Yu et al., 2024). This oversight limits the diversity and scope necessary for aligning better tool-use behaviors. Consequently, it is essential to develop an agentic RL algorithm that aligns with agent-environment interaction characteristics to fully realize the potential of LLM-based agents.

To bridge this gap, we propose **A**gentic **R**einforced **P**olicy **O**ptimization (**ARPO**), an agentic RL algorithm tailored for training a multi-turn LLM-based agent. The core principle of ARPO is to encourage the policy model to adaptively branch sampling during high-entropy tool-call rounds, thereby efficiently aligning step-level tool-use behaviors. In detail, we propose an entropy-based adaptive rollout mechanism that integrates both global and partial sampling perspectives. In the rollout phase, the LLM initially performs multiple global samplings, recording the initial entropy distribution of each sample. After each tool-calling, we further monitor the real-time token entropy variation, regarding them as branching criteria. If the entropy variation exceeds a predefined threshold, the model triggers additional partial sampling to explore alternative tool-integrated reasoning paths. This design allows ARPO to effectively expand the original sampling space while balancing global and step-level tool-use behavior learning.

To fully exploit the benefits of adaptive sampling, we introduce the advantage attribution estimation. Specifically, we explore both hard and soft advantage settings of ARPO, assigning shared advantage values to tokens along the same source reasoning path, while tokens on branched paths receive distinct values. This design encourages the model to internalize stepwise differences in tool-use effectiveness.

We conduct comprehensive evaluations across 13 datasets spanning computational reasoning, knowledge reasoning, and deep search domains. As shown in Figure 1, ARPO consistently surpasses traditional trajectory-level RL algorithms in agentic training. Remarkably, ARPO achieves this with

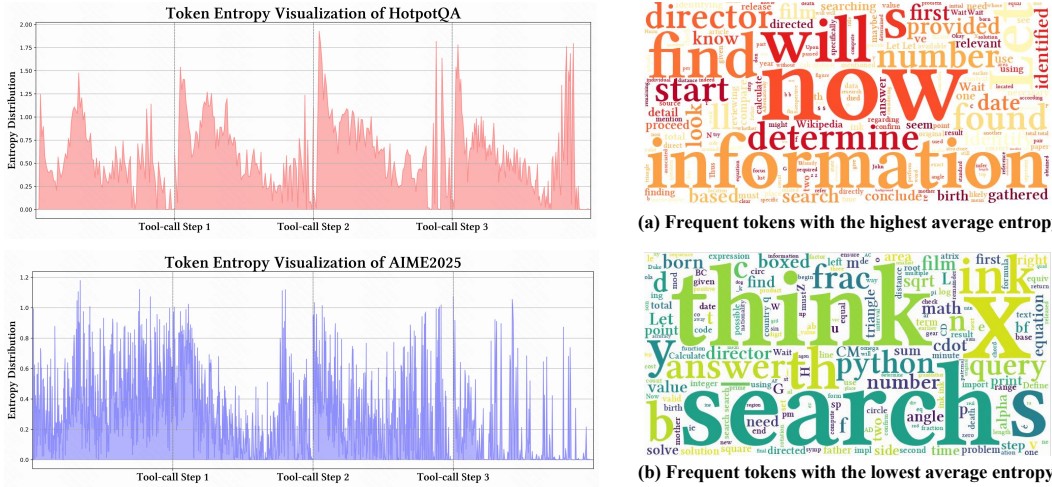

Figure 2: Token entropy variations and token frequency statistics of tool-use agents.

**only half the tool-call budget** required by other RL methods, demonstrating a compelling trade-off between accuracy and efficiency. In summary, the key contributions are as follows:

- We pioneeringly quantify the token entropy variation of LLM during agentic reasoning, revealing the inherent limitations of trajectory-level RL algorithms for aligning LLM-based agents.
- We propose the ARPO algorithm, which integrates an entropy-based adaptive rollout mechanism to balance global sampling while encouraging branch sampling during high-entropy tool-use steps. Furthermore, ARPO employs Advantage Attribution Estimation to assist the LLM in better internalizing advantage differences in stepwise tool-use behaviors.
- Beyond heuristic motivation, we also theoretically demonstrate the rationale of applying the ARPO algorithm in LLM-based agent training.
- ARPO significantly outperforms mainstream RL algorithms, requiring only half the tool-use training budgets, thereby offering practical insights for exploring agentic RL algorithms.

## 2 PRELIMINARY

Before introducing ARPO, we first review preliminary entropy-based experiments on LLM reasoning with tools. We also give a detailed preliminary of agentic reinforcement learning in Appendix F.1.

**1) Token Entropy Calculation.** Following recent entropy-based RL studies (Wang et al., 2025b;c; Cheng et al., 2025; Zheng et al., 2025b), we compute the token-level generation entropy at step $t$ as:

$$H_t = -\sum_{j=1}^{V} p_{t,j} \log p_{t,j}, \quad \text{where } \boldsymbol{p}_t = \pi_\theta \left( \cdot \mid \mathcal{R}_{<t}, x; T \right) = \text{Softmax} \left( \frac{\boldsymbol{z_t}}{\tau} \right). \quad (1)$$

Here, $V$ is the vocabulary size, $\boldsymbol{z_t} \in \mathbb{R}^V$ is the pre-softmax logits, and $\tau$ is the decoding temperature. Note that this entropy reflects the uncertainty in the token generation distribution

**2) Pilot Experiment on Token Entropy.** To gain deeper insights into the reasoning process of LLM-based tool-use agents, we conduct a pilot study with two types of agents: one using a search engine for knowledge-intensive tasks and another using a Python interpreter for computational tasks. In Figure 2, we measure token entropy variations throughout the reasoning process to assess uncertainty.

Our key observations are: **(1)** Entropy rises sharply in the first 10–50 tokens following each tool call; **(2)** Entropy tends to increase during early reasoning stages, but remains lower than after receiving tool-call feedback; **(3)** Search feedback introduces more uncertainty than Python feedback.

We attribute these effects to the distributional shift between external feedback and the model's internal reasoning (**Ob.1**), which introduces uncertainty often exceeding that of the original input (**Ob.2**).

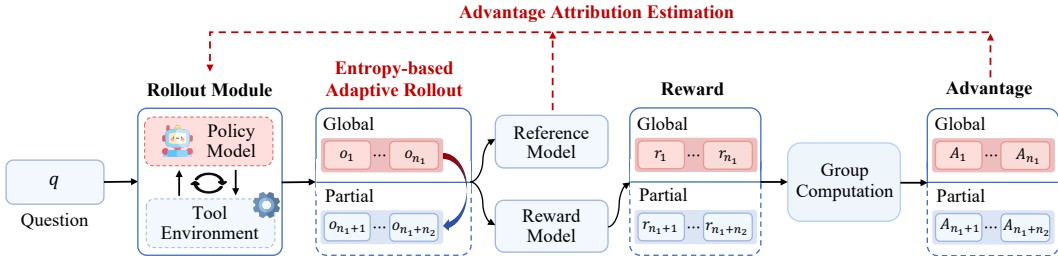

Figure 3: The overview of the ARPO algorithm.

Furthermore, search engines typically return informative textual content, whereas Python outputs consist of deterministic numbers, resulting in greater entropy fluctuations in the former case (**Ob.3**).

These findings highlight a limitation of trajectory-level RL methods, which **focus on initial reasoning while overlooking the uncertainty introduced by tool-call feedback.** Our proposed ARPO algorithm addresses this by incorporating entropy-based exploration tailored to LLM agent training.

**3) Agentic Tool Design.** In this work, we mainly focus on optimizing the training algorithms of LLM-based tool-use agents. After a comprehensive review of agentic RL studies (Dong et al., 2025; Feng et al., 2025a; Jin et al., 2025a), we identify three representative tools to evaluate the effectiveness of ARPO: **(1) Search Engine:** Retrieves relevant information by executing queries across the web. **(2) Web Browser Agent:** Accesses and parses relevant web links returned by the search engine, extracting and summarizing key content. **(3) Code Interpreter:** Automatically executes code generated by the LLM, returning execution results.

## 3 AGENTIC REINFORCE POLICY OPTIMIZATION

**Overview.** In this section, we propose the ARPO algorithm, designed to guide LLMs in exploring step-wise tool-use behaviors under entropy-based guidance, as illustrated in Figures 3 and 4:

- **Entropy-based Adaptive Rollout** (§3.1): Inspired by the entropy variations observed in preliminary experiments (§2), ARPO extends the traditional rollout process by performing not only trajectory-level sampling but also branching at high-entropy tool-use steps. By striking a balance between global and partial sampling, ARPO encourages broader exploration of tool-use behaviors.

- **Advantage Attribution Estimation** (§3.2): To better accommodate the adaptive rollout mechanism, we propose the advantage attribution estimation, enabling the model to more effectively internalize the advantage differences in stepwise tool-use behaviors.

- **Theoretical Analysis** (§3.3): To establish the theoretical foundation, we provide a formal analysis demonstrating ARPO's strong adaptability in multi-turn training scenarios for LLM-based agents.

### 3.1 ENTROPY-BASED ADAPTIVE ROLLOUT

Inspired by preliminary experiments (§2), we incorporate both trajectory-level sampling and entropy-based partial sampling during the rollout phase to cover a more comprehensive sampling scope. The design of this mechanism involves the following four core steps:

**(1) Rollout Initialization:** Given a global rollout size of $M$, the LLM first generates $N$ trajectories via trajectory-level sampling based on the input question $q$, while the remaining $M - N$ trajectories budgets are reserved for partial sampling. We then compute the entropy of the first tokens $k$ in each trajectory using Equation 1, forming the initial entropy matrix denoted as $H_{\text{initial}} \in \mathbb{R}^{1 \times k}$.

**(2) Entropy Variation Monitoring:** After recording the $H_{\text{initial}} \in \mathbb{R}^{1 \times k}$, the model performs agentic reasoning with tools, as defined in Equation 8. To continuously monitor the entropy dynamics following each tool invocation, we allow the model to generate $k$ additional tokens after concatenating the tool-call result. For the tool-call step $t$, we first compute a step-level entropy matrix denoted as $H_t \in \mathbb{R}^{1 \times k}$, then quantify the normalized change in entropy relative to the initial state as: $\Delta H_t = \text{Normalize}(H_t - H_{\text{initial}})$. Notably, the normalization means summing all the values of $\Delta H$

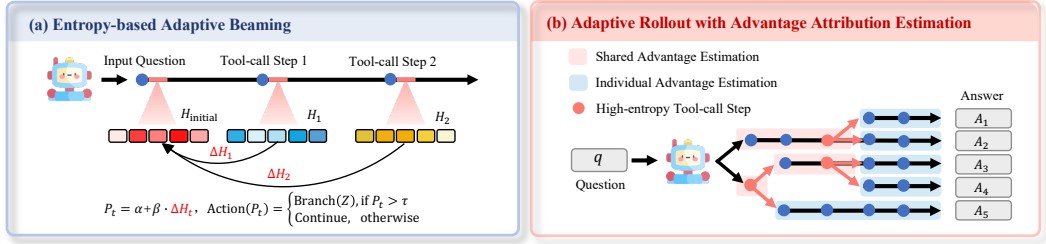

Figure 4: Illustration of ARPO's two core components. **Left:** Entropy-based adaptive beaming dynamically expands sampling based on token-level entropy. **Right:** Advantage attribution assigns shared or distinct values to tokens in inter-group reasoning paths.

and dividing by the vocab size $V$. A positive $\Delta H$ indicates an increase in uncertainty after the tool-call step $k$, whereas a negative value reflects a reduction in uncertainty.

**(3) Entropy-based Adaptive Beaming:** To encourage adaptive exploration along tool-use paths that exhibit beneficial entropy variations, we define the partial sampling probability at the step $t$ as:

$$P_t = \alpha + \beta \cdot \Delta H_t, \quad \text{Action}(P_t) = \begin{cases} \text{Branch}(Z), & \text{if } P_t > \tau; \\ \text{Continue}, & \text{otherwise.} \end{cases} \quad (2)$$

Here, $\alpha$ denotes the base sampling probability, and $\beta$ represents the stability entropy. As illustrated in Figure 4(a), the model uses $P_t$ to guide its branching behavior: if $P_t$ exceeds a threshold $\tau$, it triggers Branch($Z$) to spawn $Z$ partial reasoning paths from the current node; otherwise, it proceeds along the existing trajectory. This adaptive mechanism directs exploration toward regions of the reasoning space where rising entropy signals greater potential for informative outcomes.

**(4) Termination:** The process iterates until one of the conditions is satisfied: (1) if the total number of forked paths $\hat{Z}$ reaches the partial sampling budget $M - N$, branching stops and sampling continues until a final answer is produced; (2) if all paths terminate before reaching $M - N$, we supplement with $M - N - \hat{Z}$ additional trajectory-level samples to satisfy condition (1).

By leveraging this efficient rollout mechanism, ARPO facilitates uncertainty-aware exploration, allowing LLMs to more effectively identify step-level tool-calling behavior. Meanwhile, assuming the global expansion size and the number of tokens per trajectory are $n$, ARPO reduces the computational complexity of each rollout from the trajectory-level RL's $O(n^2)$ to between $O(n \log n)$ and $O(n^2)$ [1].

### 3.2 Advantage Attribution Estimation

Our adaptive rollout mechanism naturally produces trajectories containing both shared token segments and distinct beam paths (Figure 4), which motivates us to explore a more principled agentic RL policy update strategy. To this end, we consider the following two advantage shaping settings:

**(1) Hard Advantage Estimation:** As shown in Figure 4(b), a straightforward approach is to explicitly distinguish the shared and individual parts of each trajectory at the advantage level, thereby encouraging the model to capture step-level tool-use behaviors. Given $d$ trajectories that share certain tokens while diverging in others, we compute the advantage for the individual tokens using the normalized reward $R_i$: $\hat{A}_{i,t} = \frac{r_i - \text{mean}(\{R_i\}_{i=1}^G)}{\text{std}(\{R_i\}_{i=1}^G)}$. For the shared tokens, we assign the average advantage across $d$ trajectories that contain the shared segment: $\hat{A}_{i,t}^{\text{shared}} = \frac{1}{d} \sum_{i=1}^d \hat{A}_{i,t}$.

**(2) Soft Advantage Estimation:** An elegant alternative to hard advantage shaping is to integrate the distinction between shared and individual token segments latently during policy update. Specifically, for the input $x$, the Group Relative Policy Optimization (GRPO) (Shao et al., 2024) enables the old

---

[1]Neglecting the minor overhead from token-level entropy calculations

policy $\pi_{\text{old}}$ to generate a response set $\{y_1, y_2, \ldots, y_G\}$ and optimizes the policy by maximizing:

$$J_{\text{GRPO}}(\theta) = \mathbb{E}_{(q,a)\sim D, \{y_i\}_{i=1}^G \sim \pi_{\theta_{\text{old}}}(\cdot|q)} \left[ \frac{1}{G} \sum_{i=1}^G \frac{1}{|y_i|} \sum_{t=1}^{|y_i|} \min \left( r_{i,t}(\theta)\hat{A}_{i,t}, \right. \right.$$
$$\left. \left. \text{clip}\left(r_{i,t}(\theta), 1-\epsilon, 1+\epsilon\right)\hat{A}_{i,t}\right) - \beta D_{\text{KL}}(\pi_\theta \| \pi_{\text{ref}}) \right]. \tag{3}$$

Notably, the GRPO objective incorporates the distinction between shared and individual tokens through importance sampling ratio $r_{i,t}(\theta)$:

$$r_{i,t}(\theta) = \frac{\pi_\theta(y_{i,t} \mid x, y_{i,<t})}{\pi_{\text{old}}(y_{i,t} \mid x, y_{i,<t})}, \begin{cases} r_{i,t}(\theta) = r_{j,t}(\theta), & \text{if } y_{i,<t} = y_{j,<t} \text{ (i.e., shared tokens);} \\ r_{i,t}(\theta) \neq r_{j,t}(\theta), & \text{if } y_{i,<t} \neq y_{j,<t} \text{ (i.e., individual tokens).} \end{cases} \tag{4}$$

As indicated by Equation (4), when trajectories $y_i$ and $y_j$ undergo a partial rollout at token $t$, they share the same response prefix tokens, i.e., $y_{i,<t} = y_{j,<t}$. Consequently, the shared prefix tokens in both trajectories are assigned the same importance weight $r_{i,t}(\theta)$.

In GRPO, the mathematical interpretation is that the policy update is guided by the average advantage of tokens within each group, which serves as the loss signal.

Since shared tokens have identical $r_{i,t}(\theta)$, their advantage contributions are effectively aligned and closely approximate $\hat{A}_{i,t}^{\text{shared}}$ in a hard estimation setting. While we retain the original GRPO loss formulation, our novel partial rollout design explicitly distinguishes the update strategies between shared and individual tokens.

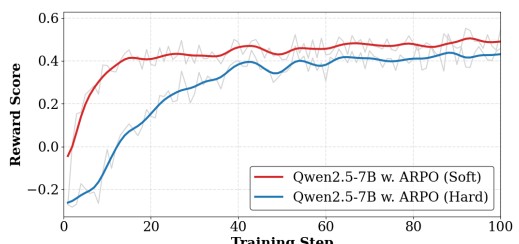

Figure 5: Comparison of different advantage estimation methods: Hard vs. Soft.

In practice, we compare reward dynamics between hard and soft advantage shaping during RL training. As shown in Figure 5, the soft setting consistently yields more stable rewards throughout ARPO training. Accordingly, ARPO adopts the soft setting as its default for advantage estimation [2].

**Hierarchical Reward Design.** The reward function serves as the optimization objective, guiding the policy model's behavior during training. We follow Tool-Star (Dong et al., 2025), considering both correctness and format rewards, along with a multi-tool collaboration reward mechanism. Notably, an additional reward $r_M$ is given when the model generates the correct answer, follows the correct tool invocation format, and uses multiple tools (i.e., `<search>` and `<python>`) during reasoning. The overall reward $R$ is formally defined as [3]:

$$R = \begin{cases} \max(Acc. + r_M, Acc.) & \text{If Format is Good \& Acc.>0;} \\ 0 & \text{If Format is Good \& Acc.=0 ;} \\ -1 & \text{Otherwise.} \end{cases} \quad r_M = \begin{cases} 0.1 & \text{If } \exists(\text{`<search>`}\&\text{`<python>`}); \\ 0 & \text{Otherwise.} \end{cases} \tag{5}$$

### 3.3 THEORETICAL FOUNDATION

Our approach leverages the adaptive partial rollout mechanism, which involves branching at high-entropy tool-use steps. Here, we elucidate the rationale behind this mechanism. In Figure 4, the adaptive partial rollout mechanism dynamically segments the Transformer-based policy's output tokens $< OT_1, OT_2, ..., OT_{|output|} >$ into $K$ segments. Each segment is defined as a macro action, $MA_i \triangleq < OT_m, OT_{m+1}, ..., OT_{m+n} >$. The corresponding macro states are defined as $MS_1 \triangleq < IT_1, IT_2, ..., IT_{|input|} >$ and $MS_i \triangleq < MS_{i-1}, MA_{i-1} >$. This segmentation allows us to derive

---

[2]The theoretical relationship between hard and soft advantage estimation is formally proven in Appendix F.2

[3]The detailed flowchart for the ARPO algorithm can be found in Algorithm 1.

the Generalized Policy Gradient (GPG) Theorem applicable to all Transformer-based policies:

$$\nabla_\theta J(\theta) = \mathbb{E}_{\tau \sim \pi_\theta} \{ \sum_{T=1}^{K} [\nabla_\theta \log \pi_\theta(MA_T | MS_T) A_T(\tau)] \}, \qquad (6)$$

where $T$ represents the macro step, and $A_T(\tau)$ denotes the advantage of trajectory $\tau$. The GPG Theorem asserts that for any differentiable Transformer-based policy $\pi_\theta$ and any objective function $J(\theta)$, optimization can be effectively conducted using macro actions (i.e., partial rollout segments). This generalization encompasses the traditional Policy Gradient Theorem (Sutton et al., 1999), $\nabla_\theta J(\theta) = \mathbb{E}_{\tau \sim \pi_\theta} \{ \sum_{t=1}^{H} [\nabla_\theta \log \pi_\theta(a_t | s_t) A_t(\tau)] \}$, which operates on single-token actions (where $a_t$ is a single output token of the Transformer), as a specific instance of our broader GPG framework. Consequently, ARPO, as an advanced implementation of the GPG Theorem, provides a robust theoretical foundation. The formal proof of the GPG Theorem is presented in Appendix F.3.

## 4 EXPERIMENTAL SETUP

**Datasets.** To comprehensively assess ARPO's effectiveness in training multi-turn tool-using agents, we conduct an evaluation on the following types of long-horizon reasoning tasks: **(1) Mathematical Reasoning:** including AIME2024, AIME2025[4], MATH500 (Lightman et al., 2024), MATH (Hendrycks et al., 2021), and GSM8K. **(2) Knowledge-Intensive Reasoning:** including HotpotQA (Yang et al., 2018), 2WikiMultihopQA (Ho et al., 2020), and Musique (Trivedi et al., 2022) and bamboogle (Press et al., 2023). **(3) Deep Search:** including General AI Assistant (GAIA) (Mialon et al., 2024), WebWalker (Wu et al., 2025a), Humanity's Last Exam (HLE) (Phan et al., 2025), and xbench-DeepSearch (Chen et al., 2025a). To ensure consistency with prior work, we adopt the Tool-Star (Dong et al., 2025) test split for math and knowledge reasoning benchmarks, and follow WebThinker and HIRA (Li et al., 2025e; Jin et al., 2025b) for the split in deep search benchmarks.

**Baselines.** To effectively evaluate the efficacy of ARPO, we consider the following three baselines: **(1) Direct Reasoning.** For 10 reasoning benchmarks, we evaluate the instruction variants of the Qwen2.5 (Qwen et al., 2024) and Llama3.1 (Dubey et al., 2024) series. Given the superior math capabilities of the Qwen3 series (Yang et al., 2025), we adopt it as the backbone for testing RL algorithms on the DeepSearch task. We also include strong reasoning baselines such as QwQ (Team, 2024b), DeepSeek-R1 (DeepSeek-AI et al., 2025), GPT-4o (Hurst et al., 2024)and o1-preview (Hurst et al., 2024). **(2) Trajectory-Level RL Algorithms.** We compare ARPO against trajectory-level RL algorithms, including GRPO (Shao et al., 2024), DAPO (Yu et al., 2025), and REINFORCE++ (Hu, 2025). **(3) LLM-Based Search Agents.** For the DeepSearch benchmark, we further benchmark ARPO against GRPO and a set of workflow-driven search agents, including vanilla RAG (Lewis et al., 2020), Search o1 (Li et al., 2025d), WebThinker (Li et al., 2025e), and ReAct (Yao et al., 2022). Detailed descriptions of all baselines are provided in Appendix D.

**Evaluation Metric** For accuracy, F1 scores are reported on four knowledge-intensive QA tasks, while others are judged by Qwen2.5-72B-instruct under the LLM-as-Judge setup. We adopt pass@1 with temperature 0.6 and top-p 0.95. Following Li et al. (2025d), answers are extracted from \box in model outputs. Detailed implementation and training guidelines are provided in Appendix E.

## 5 EXPERIMENTAL RESULTS

### 5.1 MAIN RESULTS

• **Results on Mathematical & Knowledge-Intensive Reasoning.** Our main results are shown in Table 1. In a fair setting, ARPO consistently outperforms all trajectory-level RL algorithms, firmly establishing its superiority. Moreover, we highlight the following insights:

(1) **Ineffectiveness of Prompting Methods:** Tool-integrated prompting (Li et al., 2025d) fails to elicit effective tool-use behaviors. For both Qwen and Llama series models, performance gains are marginal or even inferior to direct reasoning, indicating that prompt engineering alone is insufficient and may interfere with inherent reasoning capabilities.

---

[4]https://huggingface.co/datasets/AI-MO/aimo-validation-aime

Table 1: Overall performances on 10 challenging reasoning tasks are presented. The top two outcomes are **bolded** and underlined. Dataset abbreviations are as follows: WebW (WebWalker), HQA (HotpotQA), 2Wiki. (2wikiMultiHopQA), MuSi. (MuSiQue), and Bamb (Bamboogle).

| Method | Mathematical Reasoning | | | | | Knowledge-Intensive Reasoning | | | | | Avg. |
|---|---|---|---|---|---|---|---|---|---|---|---|
| | AIME24 | AIME25 | MATH500 | GSM8K | MATH | WebW | HQA | 2Wiki. | MuSiQ. | Bamb. | |
| **Llama3.1-8B-Instruct** | 3.3 | 0.0 | 43.3 | 81.4 | 60.6 | 3.0 | 24.3 | 24.6 | 10.4 | 40.0 | 28.8 |
| + TIR Prompting | 3.3 | 3.3 | 39.4 | 73.8 | 58.2 | 15.0 | 48.5 | 47.5 | 15.5 | 58.4 | 36.3 |
| + GRPO | 13.3 | 13.3 | 62.4 | 87.4 | 79.2 | 26.5 | 57.8 | 71.8 | 31.0 | 68.2 | 51.1 |
| + Reinforce ++ | 13.3 | **16.7** | 61.4 | 87.0 | 77.2 | 27.5 | 57.1 | 71.6 | 29.9 | 69.1 | 51.1 |
| + DAPO | 16.7 | 13.3 | 61.2 | 87.4 | 76.4 | 25.5 | 56.6 | 70.3 | 29.2 | 67.3 | 50.4 |
| + ARPO | **23.3** | **16.7** | **64.6** | **88.0** | **80.2** | **30.5** | **65.4** | **75.5** | **34.8** | **73.8** | **55.3** |
| **Qwen2.5-7B-Instruct** | 10.0 | 10.0 | 70.6 | 90.2 | 82.0 | 2.0 | 12.2 | 12.6 | 6.6 | 24.0 | 32.0 |
| + TIR Prompting | 6.7 | 10.0 | 68.2 | 64.6 | 78.2 | 15.5 | 14.8 | 18.3 | 9.5 | 23.6 | 31.0 |
| + GRPO | 23.3 | 26.7 | 78.0 | **92.8** | 87.8 | 22.0 | **59.0** | **76.1** | 30.6 | 68.4 | 56.5 |
| + Reinforce ++ | 26.7 | 23.3 | 78.0 | 92.2 | **88.8** | **26.0** | 55.1 | 68.9 | 25.2 | 64.9 | 54.9 |
| + DAPO | 20.0 | 23.3 | **80.4** | 91.0 | **88.8** | 24.0 | 57.7 | 68.4 | 28.6 | 65.5 | 54.8 |
| + ARPO | **30.0** | **30.0** | 78.8 | 92.2 | **88.8** | **26.0** | 58.8 | **76.1** | 31.1 | **71.5** | **58.3** |

(2) **Trajectory-Level RL Constraints:** Classic trajectory-level RL algorithms do not effectively harness the potential for tool-integrated reasoning compared to ARPO. While DAPO excels in single-turn tasks, it struggles in multi-turn, knowledge-intensive settings, highlighting the difficulty of inducing step-level tool-use behaviors through trajectory-level optimization.

(3) **Robust Performance of ARPO:** Under identical conditions, ARPO consistently outperforms baseline RL methods across 10 datasets, with an average accuracy gain of 4%. It generalizes well across both Qwen and Llama backbones, demonstrating strong adaptability and efficiency.

• **Results on Deep Search Tasks.** To further verify the effectiveness of our ARPO in challenging deep search scenarios, we compare the performance of the Qwen3 series models trained with only 1k RL samples, against a series of strong baseline methods. Our observations are as follows:

(1) **ARPO's Generalization in Deep Search:** In deep search settings, even state-of-the-art LLMs like GPT-4o and DeepSeek-R1-671B perform poorly on the HLE benchmark (2.0% & 8.6%). In contrast, ARPO achieves 10.0% and 43.2% pass@1 scores on HLE and GAIA using only Qwen3-14B models. Remarkably, ARPO is trained with just 1K samples from an open-source web search dataset, demonstrating strong sample efficiency and tool-use generalization.

(2) **Critical Role of Step-Level Tool Use:** ARPO consistently outperforms GRPO, with a 6% gain on GAIA and WebwalkerQA. This underscores the advantage of ARPO's balanced sampling strategy, which integrates global and step-level exploration. This design encourages diverse tool-use behaviors, particularly in high-entropy, multi-step reasoning tasks common in deep search.

## 5.2 QUANTITATIVE ANALYSIS

**Analyzing Sampling at Scale.** Due to the dynamic, multi-turn nature of Deepsearch evaluation, Pass@1 alone is insufficient to fully capture a model's tool-use potential. To address this, we conducted extended sampling analysis using Pass@3 and Pass@5 metrics, as shown in Figure 6. Both Qwen-8B and Qwen-14B exhibit consistent gains and clear scaling trends in Pass@3 and Pass@5 after ARPO alignment. Notably, Qwen-14B with ARPO achieves strong Pass@5 performance—**61.2% on GAIA, 24.0% on HLE, and 59.0% on xBench-DR**. These improvements reflect ARPO's effectiveness in promoting fine-grained tool-use exploration, thereby expanding the sampling space and enhancing both inference efficiency and behavioral diversity.[5]

**Tool-Call Efficiency Analysis.** In agentic RL training, excessive tool calls can incur substantial computational and financial costs. Therefore, an effective agentic RL algorithm must balance performance with tool-use efficiency. To evaluate ARPO's efficiency during training, we compare it

---

[5]Since xBench-DR consists entirely of Chinese queries, we use Chinese prompts for Pass@K evaluation, resulting in improved performance compared to Table 2.

Table 2: Overall performance on various deep search tasks, with accuracy results for each dataset obtained using llm-as-judge. The best results are indicated in **bold**, and the second-best results are underlined. Results from larger or closed-source models are presented in gray for reference.

| Method | General AI Assistant | | | | WebWalkerQA | | | | Humanity's Last Exam | | | | XBench |
|---|---|---|---|---|---|---|---|---|---|---|---|---|---|
| | Lv.1 | Lv.2 | Lv.3 | Avg. | Easy | Med. | Hard | Avg. | NS | CE | SF | Avg. | Avg. |
| *Direct Reasoning (>=32B)* | | | | | | | | | | | | | |
| Qwen3-32B-thinking | 26.2 | 12.1 | 0 | 14.9 | 6.9 | 1.1 | 2.9 | 3.1 | 14.6 | 9.8 | 8.4 | 12.6 | 14.0 |
| DeepSeek-R1-32B | 21.5 | 13.6 | 0.0 | 14.2 | 7.5 | 1.4 | 4.2 | 3.8 | 6.6 | 5.1 | 6.5 | 6.4 | 10.0 |
| QwQ-32B | 30.9 | 6.5 | 5.2 | 18.9 | 7.5 | 2.1 | 4.6 | 4.3 | 11.5 | 7.3 | 5.2 | 9.6 | 10.7 |
| GPT-4o | 23.1 | 15.4 | 8.3 | 17.5 | 6.7 | 6.0 | 4.2 | 5.5 | 2.7 | 1.2 | 3.2 | 2.6 | 18.0 |
| DeepSeek-R1-671B | 40.5 | 21.2 | 5.2 | 25.2 | 5.0 | 11.8 | 11.3 | 10.0 | 8.5 | 8.1 | 9.3 | 8.6 | 32.7 |
| o1-preview[†] | - | - | - | - | 11.9 | 10.4 | 7.9 | 9.9 | 12.9 | 8.1 | 6.6 | 11.1 | - |
| *Single-Enhanced Method (Qwen3-8B)* | | | | | | | | | | | | | |
| Vanilla RAG | 28.2 | 15.4 | **16.7** | 20.4 | 8.9 | 10.7 | 9.9 | 10.0 | 5.1 | 1.6 | 12.9 | 5.8 | 8.0 |
| Search-o1 | 35.9 | 15.4 | 0.0 | 21.4 | 6.7 | 15.5 | 9.7 | 11.5 | 7.6 | 2.7 | 5.3 | 6.4 | 10.0 |
| WebThinker | 43.6 | 11.5 | 0.0 | 22.3 | 6.7 | 13.1 | 16.9 | 13.0 | 7.3 | 4.0 | 6.3 | 6.6 | 13.0 |
| ReAct | 35.9 | 17.3 | 8.3 | 23.3 | 8.9 | 16.7 | 18.3 | 15.5 | 4.2 | 4.0 | 6.3 | 4.6 | 16.0 |
| *RL-based Method (Qwen3-8B)* | | | | | | | | | | | | | |
| Qwen3-8B | 28.1 | 15.4 | **16.7** | 20.4 | 0.0 | 2.4 | 2.8 | 2.0 | 3.9 | 2.7 | 8.4 | 4.6 | 9.0 |
| + GRPO | 48.7 | 25.0 | 8.3 | 32.0 | 24.4 | **33.3** | 26.8 | 29.0 | **7.9** | 4.0 | 10.5 | 7.8 | 20.0 |
| + ARPO | **53.9** | **32.7** | **16.7** | **38.8** | 26.7 | **33.3** | 29.6 | 30.5 | 7.3 | **6.7** | 15.8 | 8.8 | **25.0** |
| *Single-Enhanced Method (Qwen3-14B)* | | | | | | | | | | | | | |
| Vanilla RAG | 38.5 | 19.2 | 8.3 | 25.2 | 17.8 | 13.1 | 11.3 | 13.5 | 5.5 | 6.3 | 9.4 | 6.0 | 15.0 |
| Search-o1 | 48.7 | 23.1 | 0.0 | 30.1 | 11.1 | 21.4 | 16.9 | 17.5 | 6.4 | 4.0 | 10.5 | 6.8 | 21.0 |
| WebThinker | 48.7 | 26.9 | 8.3 | 33.0 | 13.3 | 23.8 | 18.3 | 19.5 | 7.0 | 4.0 | 9.5 | 7.0 | 23.0 |
| ReAct | 48.7 | 25.0 | 8.3 | 32.0 | 11.1 | 20.2 | 12.7 | 15.5 | 5.8 | 5.3 | 10.5 | 6.6 | 20.0 |
| *RL-based Method (Qwen3-14B)* | | | | | | | | | | | | | |
| Qwen3-14B | 33.3 | 13.5 | 0.0 | 19.4 | 6.7 | 2.4 | 4.2 | 4.0 | 5.5 | 6.7 | 11.6 | 6.8 | 14.0 |
| + GRPO | 51.3 | 34.6 | 0.0 | 36.9 | 28.9 | **33.3** | 26.8 | 30.0 | 7.9 | 6.7 | 12.6 | 8.6 | 27.0 |
| + ARPO | **56.4** | **40.4** | **16.7** | **43.7** | **31.1** | **42.9** | **31.0** | **36.0** | **10.3** | **10.7** | **13.7** | **10.0** | **32.0** |

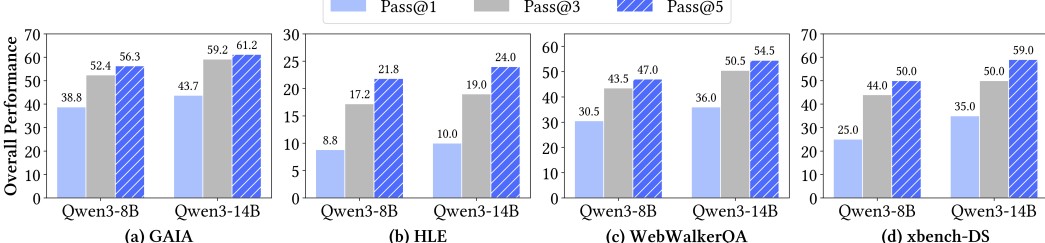

Figure 6: Analysis of Qwen3-8B and Qwen3-14B using ARPO across Pass@1 to Pass@5 metrics.

with GRPO on Qwen2.5-7B. As shown in Figure 7a, ARPO achieves higher overall accuracy while **using only half as many tool calls** as GRPO. This efficiency stems from ARPO's entropy-based adaptive rollout strategy, which selectively explores alternative branches only at high-entropy decision points. This targeted exploration significantly broadens the behavioral search space while minimizing tool usage. More ablation and scaling analyses can be found in the Appendix A.2.

**Rollout Sampling Diversity Analysis.** To demonstrate that ARPO achieves better coverage of the rollout solution space compared to GRPO, we randomly sampled 640 problems from 10 rollout steps, collecting 7.6k trajectories. Using BGEM3 as the semantic embedding model, we visualized the sampling distribution of rollouts through PCA dimensionality reduction and DBSCAN clustering.

As shown in Figure 7b, ARPO's sampling trajectories form more distinct and clearer cluster centers (54 clusters for ARPO vs. 48 for GRPO), with greater intra-cluster compactness and larger inter-cluster separation. These findings indicate that **ARPO effectively exploits the transformation of high-entropy uncertainty into exploration opportunities, significantly improving rollout diversity and making the distribution of sampled paths more structured.**

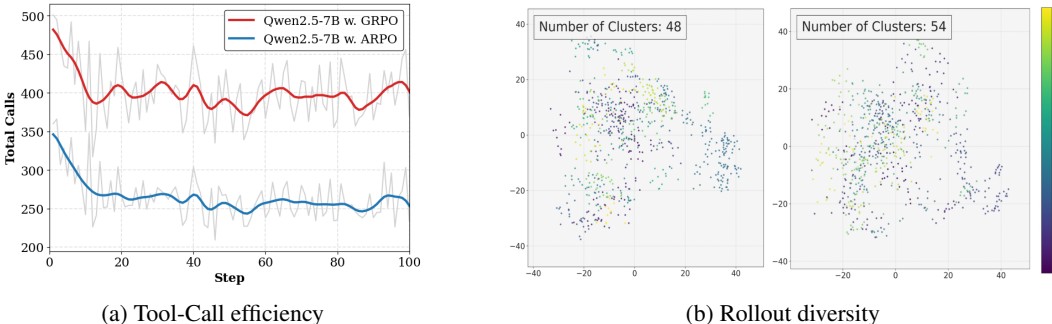

(a) Tool-Call efficiency                        (b) Rollout diversity

Figure 7: Comparison of GRPO and ARPO.

## 6  RELATED WORK

**Reinforcement Learning with Verifiable Rewards.** Reinforcement Learning with Verifiable Rewards (RLVR) (Lambert et al., 2024; Kaufmann et al., 2025) has become a prominent RLHF paradigm for LLMs, yielding substantial gains in math and code reasoning (Shao et al., 2024; DeepSeek-AI et al., 2025; Yang et al., 2025; 2024; Team, 2024b;a; Dong et al., 2024c; Qiao et al., 2024). Following OpenAI o1 (OpenAI, 2024), a growing body of work seeks to reproduce and scale RLVR recipes to broader settings (DeepSeek-AI et al., 2025; Team, 2024c; Team et al., 2025b). Recent studies improve RLVR stability through modular algorithmic and training-recipe analyses (Yu et al., 2025; Zeng et al., 2025; Hu et al., 2025; Yue et al., 2025; Feng et al., 2025b; Liu et al., 2025; Kool et al., 2019; Ahmadian et al., 2024; Dong et al., 2024a; Hu, 2025), and investigate what RLVR learns via critical-token and entropy-based perspectives (Lin et al., 2024; Gandhi et al., 2025; Li et al., 2025b; Vassoyan et al., 2025; Wang et al., 2025b; Cheng et al., 2025; Wang et al., 2025c). In parallel, segment-level RL objectives have been proposed to encourage exploration and improve credit assignment (Guo et al., 2025; Li et al., 2025g; Zheng et al., 2025a). Despite this progress, RLVR tailored to *LLM agents* remains relatively underexplored; we use entropy as a lens to study RL algorithms that more effectively shape agent behaviors.

**Agentic Reinforcement Learning.** Reinforcement learning (RL) is increasingly used to train LLM agents for interactive, open-ended environments (Lù et al., 2025; Shridhar et al., 2020; Mialon et al., 2024). Beyond classical value-based and self-play successes (Mnih et al., 2015; Silver et al., 2017; Narasimhan et al., 2015; Tan et al., 2024; Zhai et al., 2024; Bai et al., 2024; Wang et al., 2024; Schulman et al., 2017; Peng et al., 2019), recent work optimizes agent trajectories that interleave reasoning and environment interactions (Wang et al., 2025d; Zhou et al., 2024). For tool-augmented agents, many methods rely on rule-based or verifier-based rewards to encourage correct tool use, with extensions to multi-tool coordination and long-horizon settings (Jin et al., 2025a; Feng et al., 2025a; Song et al., 2025; Chen et al., 2025b; Li et al., 2025f; Sun et al., 2025a; Li et al., 2025e; Singh et al., 2025a; Qian et al., 2025a; Dong et al., 2025; Wang et al., 2025a; Li et al., 2025c). However, trajectory-level RL often provides overly coarse credit assignment for multi-turn tool-use decisions, motivating our step-level ARPO.

## 7  CONCLUSION

In this paper, we present Agentic Reinforced Policy Optimization (ARPO), an innovative reinforcement learning algorithm tailored for training multi-turn, LLM-based agents. Our experiments reveal that LLMs exhibit high token entropy after tool usage. ARPO leverages this by incorporating an entropy-based adaptive rollout mechanism, balancing global and step-level sampling to encourage diverse exploration in high-entropy tool-use phases. By integrating Advantage Attribution Estimation, ARPO enables LLMs to internalize advantage differences in stepwise tool-use interactions. Across 13 challenging benchmarks in computational reasoning, knowledge reasoning, and deep search domains, ARPO consistently outperforms traditional trajectory-level RL algorithms. Remarkably, it achieves great performance with only half the tool-use budget of other methods, offering a scalable solution for aligning LLM-based agents with dynamic environments.

ACKNOWLEDGMENTS

This work is supported by National Natural Science Foundation of China No.62272467, China Postdoctoral Science Foundation under Grant Number 2025T180440, and Outstanding Innovative Talents Cultivation Funded Programs 2026 of Renmin University of China. The work is partially done at the Beijing Key Laboratory of Research on Large Models and Intelligent Governance.

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

## Appendix

# A  SUPPLEMENT EXPERIMENTAL RESULTS

## A.1  ABLATIONS OF BROWSER AGENTS.

To further assess the impact of the browser agent on the deepsearch task, we design three experimental settings with progressively stronger browsing capabilities: **(1)** no browser, where only retrieved snippets are available; **(2)** a browser agent of comparable scale to the reasoning model; and **(3)** a larger-scale browser agent with more parameters.

As shown in Table 3, results show that the setting without a browser exhibits the worst performance consistency, indicating that relying solely on rule-generated web snippet summaries is insufficient to provide the necessary information support in deep search tasks. This highlights the necessity of web content fetching and browsing. As the capability of the browser agent increases, model performance also improves significantly, demonstrating that a more powerful search agent can more effectively integrate information and extract key details relevant to the question. In summary, the capability of the external browser agent is highly correlated with the accuracy of the Deepsearch task and shows a clear upward trend as its scale increases.

## A.2  SCALING ANALYSIS OF ARPO

To verify the scalability of ARPO and gain deeper insights into its characteristics, we use the Qwen2.5-7B model as the backbone for a scaling analysis of three core parameters: *entropy value*, *global rollout size*, and *initial sampling size*. Our observations are as follows:

**Entropy Value** ($\Delta H_t$)**:** As shown in Figure 8 (left), model performance increases with rising entropy values, peaking at 0.4. This indicates that integrating a moderate amount of entropy as a clue for partial sampling substantially enhances the model's ability to explore rare tool-use behaviors, thereby improving training outcomes. However, as entropy reaches 1.0, perfor-

Table 3: Ablation studies of the backbone model of browser agents in deep search tasks.

| Method | GAIA | HLE | WebWalk. | Avg. |
|---|---|---|---|---|
| ***Qwen3-8B*** | | | | |
| + Snippet only | 33.0 | 7.5 | 29.0 | 23.2 |
| + Qwen3-8B Browser | 38.8 | **8.8** | 30.5 | 26.0 |
| + QWQ-32B Browser | **38.8** | 8.2 | **33.0** | **26.6** |
| ***Qwen3-14B*** | | | | |
| + Snippet only | 35.0 | 8.4 | 31.0 | 24.8 |
| + Qwen3-14B Browser | 43.7 | 10.0 | 36.0 | 29.9 |
| + QWQ-32B Browser | **47.6** | **32.3** | **38.4** | 39.4 |

mance declines, suggesting a trade-off in the weight of entropy in sampling. Over-reliance on entropy may reduce sampling diversity, confirming the necessity of balancing base sampling probabilities $\alpha$ with entropy in ARPO.

**Initial Sampling Size** ($N$)**:** Figure 8 (middle) illustrates that as the initial sampling size increases, model performance improves, peaking at 8. Notably, with a global rollout size of 16, increasing the initial sampling size from 0 to 8 shifts the global-to-partial sampling ratio from 1:15 to 1:1. This underscores the importance of balancing sampling proportions for improving performance. As anticipated, increasing the size to 16 results in a great performance decline. This is because it leads to complete global sampling, which disrupts the dynamic sampling balance.

**Global Rollout Size** ($M$)**:** As depicted in the Figure 8 (right), increasing the global rollout size enhances model performance, indicating that the ARPO algorithm is scalable and can improve generalization performance with larger sizes.

# B  SUPPLEMENT RELATED WORK

## B.1  REINFORCEMENT LEARNING WITH VERIFIABLE REWARD.

Recently, Reinforcement Learning with Verifiable Rewards (RLVR) (Lambert et al., 2024; Kaufmann et al., 2025) has become a leading approach in Reinforcement Learning through Human Feedback (RLHF), particularly excelling in enhancing mathematical and programming reasoning (Shao et al., 2024; DeepSeek-AI et al., 2025; Yang et al., 2025; 2024; Team, 2024b;a; Dong et al., 2024c; Qiao et al., 2024). OpenAI o1 (OpenAI, 2024) first showcased RL's effectiveness in large-scale reasoning tasks. Building on this, models like DeepSeek R1 (DeepSeek-AI et al., 2025), QwQ (Team, 2024c),

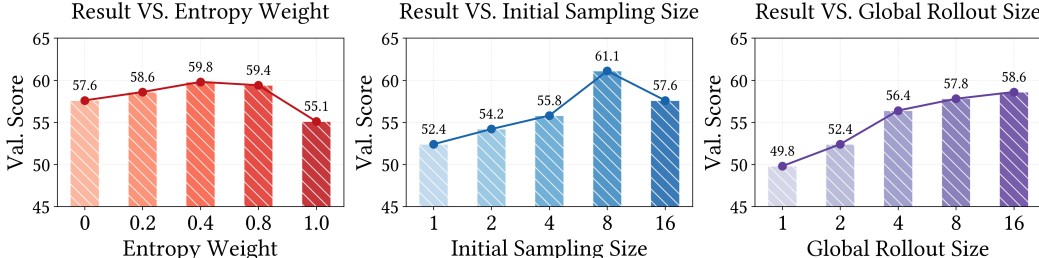

Figure 8: Scaling analysis of different Hyper-parameters in Qwen2.5-7B with ARPO. The detailed setting can be found in Appendix E.5.

and Kimi k1.5 (Team et al., 2025b) aim to replicate and surpass its performance. To improve RL algorithms' performance and stability, researchers have developed models like DAPO (Yu et al., 2025) and SimpleRLZoo (Zeng et al., 2025), exploring algorithm design across various RL modules(Hu et al., 2025; Yue et al., 2025; Feng et al., 2025b; Liu et al., 2025; Kool et al., 2019; Ahmadian et al., 2024; Dong et al., 2024a; Hu, 2025). Lin et al. identified key tokens affecting errors and showed that replacing them can alter model behavior. Studies (Gandhi et al., 2025; Li et al., 2025b) found RLVR primarily learns format over content, while several works (Vassoyan et al., 2025; Wang et al., 2025b; Cheng et al., 2025; Wang et al., 2025c) pointed out key tokens to high-entropy tokens to explore RL learning's essence. In recent work, segment-level reinforcement learning approaches have been proposed to broaden the exploration space (Guo et al., 2025; Li et al., 2025g; Zheng et al., 2025a). However, RLVR algorithms specifically for LLM agents remain underexplored. This paper uses entropy as a criterion to investigate reinforcement learning algorithms suited for LLM agent behavior.

### B.2 AGENTIC REINFORCEMENT LEARNING.

Reinforcement learning (RL) is essential for enabling LLM agents to adapt to dynamic and open environments (Lù et al., 2025; Shridhar et al., 2020; Mialon et al., 2024). Foundational works like DQN (Mnih et al., 2015) and AlphaZero (Silver et al., 2017) demonstrate that self-play-based RL can equip agents with capabilities from natural language understanding to strategic gameplay (Narasimhan et al., 2015). Building on this, value-based RL approaches have been employed to enhance embodied intelligence in hardware control and complex gaming tasks (Tan et al., 2024; Zhai et al., 2024; Bai et al., 2024; Wang et al., 2024; Schulman et al., 2017; Peng et al., 2019). Recent efforts, exemplified by RAGEN (Wang et al., 2025d; Zhou et al., 2024), integrates reasoning states and environmental interactions into turn-level responses using trajectory-level RL. To improve tool-integrated reasoning, studies (Jin et al., 2025a; Feng et al., 2025a; Song et al., 2025; Jin et al., 2025a; Chen et al., 2025b; Feng et al., 2025a; Li et al., 2025f; Sun et al., 2025a; Li et al., 2025e; Singh et al., 2025a) employ rule-based RL to teach LLMs how to autonomously invoke external tools (e.g. search engines, Python compilers) to boost reasoning accuracy. Further research, including ToolRL (Qian et al., 2025a), Tool-Star (Dong et al., 2025), and OTC (Wang et al., 2025a) explores multi-tool integration and tool-use efficiency. A series of works led by Kimi Deepresearcher [6] and Websailor (Li et al., 2025c) optimize RL algorithms to better adapt to deepsearch's long context scenarios. While most works improve tool invocation through reward shaping and rollout mechanisms, simply applying trajectory-level RL fails to effectively capture the multi-turn, long-horizon characteristics of LLM-based agent behavior. This motivates the proposal of ARPO to attempt learning step-level tool-use behavior patterns.

## C DATASETS

In this section, we provide a detailed introduction to the datasets used in ARPO's experiments.

### C.1 MATHEMATICAL REASONING BENCHMARKS

- **AIME24** [7] is a dataset in evaluating the mathematical reasoning ability of models. It consists of 30 challenging math problems. All of them are from the American Invitational Mathematics

---

[6] https://moonshotai.github.io/Kimi-Researcher/
[7] https://huggingface.co/datasets/HuggingFaceH4/aime_2024

Examination. The problems in the AIME24 dataset cover a wide variety of mathematical fields such as algebraic equations and geometric puzzles. Due to the difficulty characteristics and the richness of question types, it has become a popular benchmark for evaluating the reasoning performance of models, and is widely used in multiple related research experiments.

- **AIME25** [8] consists of 30 challenging math problems. It is directly composed of the real questions from the American Invitational Mathematics Examination (AIME I & II) newly released in February 2025. AIME25's knowledge areas are extremely wide. It deeply covers core mathematical sections such as algebra, geometry, number theory, and combinatorial mathematics. This characteristic enables the AIME25 dataset to effectively distinguish the mathematical reasoning abilities of different models.

- **MATH500 (Lightman et al., 2024)** is selected by OpenAI from the MATH evaluation dataset. It contains 500 high-difficulty math problems. These problems cover multiple mathematical fields such as algebra, geometry, calculus, and number theory. The difficulty is close to or exceeds the college level. In academic research, MATH500 dataset is often used to evaluate the performance of various reasoning models.

- **MATH (Hendrycks et al., 2021)** is a significant academic dataset. It is designed to test and enhance models' mathematical reasoning skills. It covers a wide range of mathematical fields, including abstract algebra, calculus, and discrete mathematics. The dataset divides training data into three levels, which helps effectively evaluate model performance at different stages.

- **GSM8K (Cobbe et al., 2021)** is an elementary school math problem dataset released by OpenAI. These problems require 2 to 8 steps to solve, mainly through a series of basic calculations to obtain the final answer. This dataset is primarily used to test the logical and mathematical abilities of models and has been applied in multiple benchmark tests.

## C.2 Knowledge-Intensive reasoning benchmarks

- **HotPotQA (Yang et al., 2018)** is a question-answering dataset for multi-hop. All the documents are sourced from Wikipedia, which provides the dataset with a rich knowledge base and relatively structured information. It is an important benchmark for evaluating the ability of LLMs to understand complex search tasks.

- **2WikiMultihopQA (Ho et al., 2020)** is a dataset specifically designed for the multi-hop question-answering task. It aims to test and evaluate the ability of natural language processing models to answer questions that require multi-step reasoning and the integration of information from different documents.

- **Musique (Trivedi et al., 2022)** is a question-answering dataset specifically designed for the multi-hop question-answering task. Musique aims to be a challenging benchmark for evaluating models' multi-hop question-answering ability. It promotes the development of models from simple information retrieval to deeper semantic understanding and logical reasoning.

## C.3 Deep Search Benchmarks

- **GAIA (Mialon et al., 2024)** is designed to evaluate the comprehensive capabilities of LLMs in real-world tasks. The dataset contains 466 carefully designed questions to test the performance of AI systems in basic capabilities. It contains tasks such as reasoning, web browsing, and tool use. The proposal of GAIA provides a new framework for the evaluation of general artificial intelligence assistants.

- **HLE (Phan et al., 2025)** is an emerging and highly challenging benchmark dataset. It aims to deeply evaluate the performance of LLMs when faced with complex questions requiring deep understanding and complex reasoning. This dataset covers a large number of marginal, interdisciplinary problems that demand highly abstract thinking to solve. Different from traditional benchmarks, HLE aims to simulate an ultimate test of AI intelligence.

- **WebWalker (Wu et al., 2025a)** is a dataset used to evaluate the performance of LLMs in web traversal tasks. This dataset contains 680 question-answer pairs. It aims to address the limitations of LLMs when dealing with complex information. Additionally, it improves the models' capabilities in multi-hop reasoning and dynamic web page structures.

---

[8] https://huggingface.co/datasets/math-ai/aime25

- **xbench-DeepSearch (Chen et al., 2025a)** is an evaluation set for assessing the deep search capabilities of AI agents. This dataset takes full consideration of the breadth of the search space and the depth of reasoning. Different from existing knowledge search benchmarks, xbench-DeepSearch is more capable of examining the high-order capabilities of agents.

## D BASELINES

In this section, we introduce baselines used in our work.

### D.1 DIRECT REASONING

- **Qwen2.5 Series (Qwen et al., 2024)** is a series of LLMs developed by the Alibaba team. It includes the general-purpose language model Qwen2.5, the programming-specific model Qwen2.5-Coder, and the mathematics-specific model Qwen2.5-Math. The Qwen2.5 series of models have been pretrained on large-scale datasets. Compared with past Qwen series of models, the Qwen2.5 series have a richer knowledge reserve. In addition, it has good performance in various tasks such as programming, mathematics, and instruction following.

- **Llama3.1 Series (Dubey et al., 2024)** is a series of natural language generation model launched by Meta. It includes three specifications: 8B, 70B, and 405B. These models can handle longer text inputs and generate more coherent long-text outputs. This series of models also performs well in multilingual tasks. The Llama 3.1 series of models have undergone performance tests on more than 150 benchmark datasets. Its large-scale model is competitive with leading base models in a series of tasks. The smaller 8B and 70B models also perform well in comparisons with closed-source and open-source models with a similar number of parameters.

- **Qwen 3 Series (Yang et al., 2025)** is a series of open-source model developed by Alibaba. The Qwen3 series of models includes 2 Mixture-of-Experts (MoE) models and 6 Dense models, with the number of parameters ranging from 0.6B to 235B. Qwen3 natively supports the thinking mode and non-thinking mode. In the thinking mode, the model reasons step by step and is suitable for handling complex problems. The non-thinking mode can provide a fast, nearly instant response and is suitable for simple problems. Qwen3 builds a training corpus based on approximately 36 trillion tokens, ensuring the model's powerful capabilities and flexibility.

- **QwQ (Team, 2024b)** is an open-source inference model launched by the Alibaba team. It focuses on enhancing AI's capabilities in mathematics, programming, and complex logical reasoning. The QwQ-32B is a dense model with 32 billion parameters. It surpasses most existing models in core tasks such as mathematical reasoning and code-generation ability. The QwQ-32B achieves breakthroughs through innovative multi-stage reinforcement learning. Its core training approach lies in gradually expanding general capabilities while consolidating specialized advantages.

- **DeepSeek-R1 (DeepSeek-AI et al., 2025)** is a reasoning model developed by DeepSeek-AI. DeepSeek-R1 is trained using reinforcement learning. The inference process involves a large amount of reflection and verification, and the length of the thought chain can reach tens of thousands of tokens. It performs outstandingly in tasks such as mathematics, code, and various complex logical reasoning.

- **GPT-4o (Hurst et al., 2024)** is a multimodal LLM released by OpenAI. GPT-4o can accept any combination of text, audio, and images as input. In addition, it can generate any combination of text, audio, and images as output. GPT-4o has achieved performance comparable to that of GPT-4 Turbo in aspects such as text, reasoning, and coding. Moreover, it has also set new highs in performance scores for multilingual, audio, and visual functions.

- **o1-preview (Hurst et al., 2024)** is a preview version model in the o1 series of LLMs launched by OpenAI. It represents an important breakthrough in the field of reasoning. o1-preview is based on the GPT-4 architecture and trained through reinforcement learning. It aims to enhance the reasoning ability for complex tasks and the ability to solve practical problems. It can exhibit powerful abilities in tasks that require in-depth reasoning.

### D.2 TRAJECTORY-LEVEL RL ALGORITHMS

- **GRPO (Shao et al., 2024)** is a reinforcement learning algorithm based on policy optimization. It aims to address the balance issue among stability, sample efficiency, and theoretical guarantees

in traditional policy optimization methods. By introducing the concept of relative advantage, it simplifies the calculation while maintaining the theoretical guarantee of policy improvement. The GRPO algorithm is applicable to reinforcement learning tasks in both continuous and discrete action spaces.

- **DAPO (Yu et al., 2025)** is a LLM reinforcement learning algorithm developed by ByteDance Labs. It aims to address the key challenges of large-scale RL training. It performs outstandingly in complex tasks such as mathematical reasoning and code generation. The Clip-Higher strategy proposed by DAPO effectively increases the entropy value, facilitating the generation of more diverse samples. In addition, it introduces mechanisms such as dynamic sampling, Token-Level Policy Gradient Loss calculation, and Overlong Reward Shaping to stabilize the training process.

- **REINFORCE++ (Hu, 2025)** is a new algorithm for improved versions of the classic REINFORCE algorithm. Its core objective is to address the limitations of the original REINFORCE, and enhance performance by integrating multiple optimization strategies. REINFORCE++ typically incorporates a baseline function to reduce variance by subtracting the baseline. Through the baseline and TD estimation, REINFORCE++ makes the gradient update more stable. It doesn't need to wait for a complete trajectory and supports incremental updates. In addition, it avoids premature policy rigidity through entropy regularization.

### D.3 LLM-BASED SEARCH AGENT

- **RAG (Lewis et al., 2020)** (Retrieval-Augmented Generation) is a technical approach that combines information retrieval with a generative model. It aims to improve the accuracy, reliability, and timeliness of the output of generative models. Its core idea is: before generating an answer, first retrieve information related to the question from an external knowledge base, and then let the model generate a response based on the retrieved content. This can solve the problem of internal knowledge deficiency or hallucination within the model to some extent.

- **Search-o1 (Li et al., 2025d)** is an Agentic search-enhanced reasoning model framework. It is mainly designed to address the knowledge deficiency problem existing in the reasoning process. By integrating the Agentic RAG mechanism and the Reason-in-Documents module, it improves the accuracy, coherence, and reliability of model reasoning. Experiments show that Search-o1 outperforms native reasoning and traditional RAG methods in complex reasoning tasks.

- **WebThinker (Li et al., 2025e)** is an open-source in-depth research framework launched by Renmin University of China. It endows LRMs with the ability to autonomously search, deeply explore web pages, and write research reports. WebThinker has developed a training strategy based on direct preference optimization. It uses training with preference data through iterative synthesis tools to enhance the tool utilization ability of LRMs.

- **ReAct (Yao et al., 2022)** is an artificial intelligence method that combines reasoning and acting. It aims to enable models to solve complex tasks more effectively through a "thinking while doing" mode similar to human thinking. Its core idea is to break the limitation of the traditional model, allowing the model to actively generate reasoning steps and call external tools (such as search engines, databases, etc.) during the decision-making process, and finally obtain the answer through iterative optimization.

## E  IMPLEMENTATION DETAILS

### E.1  TRAINING GUIDELINE

Our study aims to **validate the effectiveness of ARPO at the algorithmic level compared to traditional RL in training LLM agents, rather than merely pursuing performance improvements.** To ensure reproducibility, all training frameworks and datasets are sourced from open-access resources. Specifically, our experiments adhere to the cold-start SFT with RL paradigm (Song et al., 2025; Dong et al., 2025) to mitigate reward collapse during the initial RL training phases.

**1. Cold-Start Finetuning Phase:** Utilizing the LLaMAFactory (Zheng et al., 2024) framework, we leverage Tool-Star's open-source dataset of 54K training samples. To enrich the quality of mathematical reasoning data, we incorporate the STILL dataset (0.8K), drawing inspiration from CORT (Li et al., 2025a).

**2. RL Phase:** To assess ARPO across various scenarios, we explore the following domains:

- **Deep Reasoning Tasks:** This includes computational reasoning (e.g., AIME24, MATH500) and multi-hop knowledge-based reasoning (e.g., HotpotQA, Bamboogle). We utilize Tool-Star's 10K open-source RL training samples for algorithmic comparison.

- **Deep Search Tasks:** These tasks require extensive web exploration and information integration, necessitating longer contexts and frequent tool interactions. We use **only 1K mixed hard search samples** from SimpleDeepSearcher (Sun et al., 2025b) and WebSailor (Li et al., 2025c) for training.

To expedite the RL training phase, we incorporate top-10 snippets from the Bing search engine as search results, employ a Python compiler within a sandbox environment, and use token-level F1 scores as the correctness signal [9].

### E.2 SUPERVISED FINE-TUNING

As mentioned in Section E.1, during the supervised fine-tuning phase, we train the Qwen2.5-3B-Instruct model using the Llama Factory framework with a learning rate of $7 \times 10^{-6}$. We employ DeepSpeed ZeRO-3 (Rasley et al., 2020) and FlashAttention2 (Dao, 2023) for optimization. The batch size is set to 128, with a weight decay of 0.1, and the model is trained for 3 epochs. We use BF16 mixed precision with a maximum input length of 4096 tokens.

### E.3 REINFORCEMENT LEARNING

In the ARPO phase, we implement the ARPO algorithm based on the VERL framework (Sheng et al., 2024). Notably, all tool invocation results are excluded from loss calculation to prevent bias towards tool outputs. The loss computation only considers tokens involved in text reasoning and tool requests. We differentiate settings for Deep Reasoning Tasks and Deep Search Tasks:

**1. Deep Reasoning Tasks:** For models with 7B parameters, whether using ARPO or other trajectory-level RL methods, our standard setup includes a total training batch size of 128, a PPO mini-batch size of 16, a global rollout size of 16, and an initial sampling size of 8. Each interaction response length is capped at 4096 tokens. For ARPO rollouts, we set the entropy weight to 0.2, the parameter $a$ to 0.5, and the threshold to 0.5. To stabilize training, the KL divergence coefficient in GRPO is set to 0. The reinforcement learning phase spans 2 epochs, conducted on 8 NVIDIA H800 GPUs.

**2. Deep Search Tasks:** For models with 8B parameters, we maintain the same settings as in the Deep Reasoning Tasks, except that each interaction response length is extended to 8192 tokens. For 14B models, the same parameters are used, but experiments are conducted on 16 NVIDIA H800 GPUs. Due to a limited dataset of 1K samples, the reinforcement learning phase lasts for 5 epochs.

### E.4 DETAILS OF SEARCH

During the training and testing phases, we used the Bing Web Search API as the retriever, configured with the US-English (US-EN) locale. Following a series of related works on RAG (Jin et al., 2024; Li et al., 2024b; Dong et al., 2024e;b;d), we retrieved 10 web pages as supporting documents for each query.

For mathematical and knowledge reasoning, we evaluated using only the top 10 snippets. However, for deep search tasks, we fetched each page with up to 6000 tokens from the URLs and used a model of the same size as the reasoning model as a browser agent to refine the information.

### E.5 SCALING EXPERIMENT SETUP

In our scaling experiments, we align with the aforementioned settings: a total training batch size of 128, a PPO mini-batch size of 16, a global rollout size of 16, and an initial sampling size of 8. For ARPO rollouts, the entropy weight is 0.2, $a$ is 0.5, and the threshold is 0.5. We vary specific parameters for targeted experiments while keeping others constant.

---

[9]In this paper, we only use the LLM to correct grammatical errors.

## F  Theoretical Analysis and Proofs

### F.1  Preliminary of Agentic Reinforcement Learning

In this section, we formulate the agentic RL training objective as:

$$\max_{\pi_\theta} \mathbb{E}_{x \sim \mathcal{D}, y \sim \pi_\theta(\cdot|x;T)} \left[ r_\phi(x, y) \right] - \beta \, \mathbb{D}_{\mathrm{KL}} \left[ \pi_\theta(y \mid x; T) \, \| \, \pi_{\mathrm{ref}}(y \mid x; T) \right], \tag{7}$$

where $T$ denotes the set of available tools, $\pi_\theta$ represents the policy LLM, $\pi_{\mathrm{ref}}$ is the reference LLM, $r_\phi$ and $\mathbb{D}_{\mathrm{KL}}$ denotes the reward function and KL divergence respectively. The input $x$ is sampled from dataset $\mathcal{D}$, and $y$ is the corresponding output, possibly interleaved with tool-call feedback.

Unlike conventional RL methods that rely solely on LLM rollouts, agentic RL incorporates tool-call feedback during the reasoning process (Chen et al., 2023; Gou et al., 2024; Li et al., 2025f; Wu et al., 2025b; Li et al., 2024a). The rollout sampling can be decomposed as:

$$P_\theta(\mathcal{R}, y \mid x; T) = \underbrace{\prod_{t=1}^{t_{\mathcal{R}}} P_\theta(\mathcal{R}_t \mid \mathcal{R}_{<t}, x; T)}_{\text{Agentic Reasoning}} \cdot \underbrace{\prod_{t=1}^{t_y} P_\theta(y_t \mid y_{<t}, \mathcal{R}, x; T)}_{\text{Answer Generation}}, \tag{8}$$

where $\mathcal{R}$ is the reasoning trajectory of length $t_{\mathcal{R}}$, interleaved with tool-call feedback, and $y$ is the final answer with length $t_y$. Our ARPO is built upon rule-based RL algorithm (e.g. GRPO (Shao et al., 2024), Reinforce++ (Hu, 2025)) designed to optimize LLM-based agents.

### F.2  Theoretical Analysis of Soft Advantage Estimation

In this section, we conduct a detailed theoretical analysis of Soft Advantage Estimation. First, we present the classic GRPO optimization objective:

$$J_{\mathrm{GRPO}}(\theta) = \mathbb{E}_{(q,a) \sim D, \{o_i\}_{i=1}^G \sim \pi_{\theta_{\mathrm{old}}}(\cdot|q)} \left[ \frac{1}{G} \sum_{i=1}^G \frac{1}{|o_i|} \sum_{t=1}^{|o_i|} \min \left( r_{i,t}(\theta) \hat{A}_{i,t}, \right. \right.$$
$$\left. \left. \mathrm{clip}\left( r_{i,t}(\theta), 1 - \epsilon, 1 + \epsilon \right) \hat{A}_{i,t} \right) - \beta D_{\mathrm{KL}}(\pi_\theta \, \| \, \pi_{\mathrm{ref}}) \right] \tag{9}$$

For each individual problem, we define the optimization objective as:

$$J_{\mathrm{GRPO}}^q(\theta) = \frac{1}{G} \sum_{i=1}^G \frac{1}{|o_i|} \sum_{t=1}^{|o_i|} \min \left( r_{i,t}(\theta), \mathrm{clip}\left( r_{i,t}(\theta), 1 \pm \epsilon \right) \right) \hat{A}_{i,t} - \beta D_{\mathrm{KL}}(\pi_\theta \, \| \, \pi_{\mathrm{ref}}) \tag{10}$$

Therefore, the classical GRPO optimization objective can be expressed as:

$$J_{\mathrm{GRPO}}(\theta) = \mathbb{E}_{(q,a) \sim D, \{o_i\}_{i=1}^G \sim \pi_{\theta_{\mathrm{old}}}(\cdot|q)} [J_{\mathrm{GRPO}}^q(\theta)] \tag{11}$$

Subsequently, we focus on analyzing $J_{\mathrm{GRPO}}^q(\theta)$. Assume that for the inference of problem $q$, the partial rollout operation starts from the $l$-th token. We define two importance sampling ratios:

$$r_{i,t}(\theta)^{<l} = \frac{\pi_\theta(y_{i,t} \mid x, y_{i,<t})}{\pi_{\mathrm{ref}}(y_{i,t} \mid x, y_{i,<t})}, \tag{12}$$

$$r_{i,t}(\theta)^{>l} = \frac{\pi_\theta(p \mid x, q)}{\pi_{\mathrm{ref}}(p \mid x, q)} \frac{\pi_\theta(y_{i,t} \mid x, q, p, y_{i,<t})}{\pi_{\mathrm{ref}}(y_{i,t} \mid x, q, p, y_{i,<t})}, \tag{13}$$

where $r_{i,t}(\theta)^{<l}$ and $r_{i,t}(\theta)^{>l}$ represent the importance sampling ratios before and after the $l$-th token, respectively, $q$ represents the input question, $p$ represents shared tokens, and $y_{i,<t}$ in 13 represents the sequence from shared tokens to before the $t$-th token. In addition, we define $o_l^i$ as the $l$-th token

of the $i$-th sequence. The absolute value denotes the length of the sequence, where $|o_{l:i}|$ represents the length of the sequence from the $l$-th token to the $i$-th token. Then $J_{\text{GRPO}}^q(\theta)$ can be expressed as:

$$
\begin{aligned}
J_{\text{GRPO}}(\theta) &= \frac{1}{G}\sum_{i=1}^{G}\frac{1}{|o_i|}\left[\sum_{t=1}^{|o_l^i|}\min\left(r_{i,t}(\theta)^{<l},\text{clip}\left(r_{i,t}^{\leq l}(\theta),1\pm\epsilon\right)\right)\hat{A}_{i,t}\right.\\
&\qquad\qquad\left.+\sum_{t=|o_l^i|}^{|o_i|}\min\left(r_{i,t}^{>l}(\theta),\text{clip}\left(r_{i,t}^{>l}(\theta),1\pm\epsilon\right)\right)\hat{A}_{i,t}\right]-\beta D_{\text{KL}}(\pi_\theta\|\pi_{\text{ref}})\\[2mm]
&= \frac{1}{G}\sum_{i=1}^{G}\frac{1}{|o_i|}\left[\sum_{t=1}^{|o_l^i|}\min\left(r_{i,t}(\theta)^{<l},\text{clip}\left(r_{i,t}^{\leq l}(\theta),1\pm\epsilon\right)\right)\hat{A}_{i,t}\right]\\
&\qquad+\frac{1}{G}\sum_{i=1}^{G}\frac{1}{|o_i|}\left[\sum_{t=|o_l^i|}^{|o_i|}\min\left(r_{i,t}^{>l}(\theta),\text{clip}\left(r_{i,t}^{>l}(\theta),1\pm\epsilon\right)\right)\hat{A}_{i,t}\right]-\beta D_{\text{KL}}(\pi_\theta\|\pi_{\text{ref}})\\[2mm]
&= \frac{1}{G}\sum_{i=1}^{G}\frac{1}{|o_i|}\left[\sum_{t=1}^{|o_l^i|}\min\left(r_{i,t}(\theta)^{<l},\text{clip}\left(r_{i,t}^{\leq l}(\theta),1\pm\epsilon\right)\right)\hat{A}_{i,t}\right]\\
&\qquad+\frac{1}{G}\sum_{i=1}^{G}\frac{|o_{l:i}|}{|o_i|}\cdot\frac{1}{|o_{l:i}|}\left[\sum_{t=|o_l^i|}^{|o_i|}\min\left(r_{i,t}^{>l}(\theta),\text{clip}\left(r_{i,t}^{>l}(\theta),1\pm\epsilon\right)\right)\hat{A}_{i,t}\right]-\beta D_{\text{KL}}(\pi_\theta\|\pi_{\text{ref}})\\[2mm]
&= \frac{1}{G}\sum_{i=1}^{G}\frac{1}{|o_i|}\left[\sum_{t=1}^{|o_l^i|}\min\left(r_{i,t}(\theta)^{<l},\text{clip}\left(r_{i,t}^{\leq l}(\theta),1\pm\epsilon\right)\right)\hat{A}_{i,t}\right]\\
&\qquad+\frac{1}{G}\sum_{i=1}^{G}\frac{|o_i|-|o_l|}{|o_i|\cdot|o_{l:i}|}\left[\sum_{t=|o_l^i|}^{|o_i|}\min\left(r_{i,t}^{>l}(\theta),\text{clip}\left(r_{i,t}^{>l}(\theta),1\pm\epsilon\right)\right)\hat{A}_{i,t}\right]-\beta D_{\text{KL}}(\pi_\theta\|\pi_{\text{ref}})\\[2mm]
&= \frac{1}{G}\sum_{i=1}^{G}\frac{1}{|o_i|}\left[\sum_{t=1}^{|o_l^i|}\min\left(r_{i,t}(\theta)^{<l},\text{clip}\left(r_{i,t}^{\leq l}(\theta),1\pm\epsilon\right)\right)\hat{A}_{i,t}\right]\\
&\qquad+\frac{1}{G}\sum_{i=1}^{G}\frac{1}{|o_{l:i}|}\left[\sum_{t=|o_l^i|}^{|o_i|}\min\left(r_{i,t}^{>l}(\theta),\text{clip}\left(r_{i,t}^{>l}(\theta),1\pm\epsilon\right)\right)\hat{A}_{i,t}\right]\\
&\qquad-\frac{1}{G}\sum_{i=1}^{G}\frac{|o_l|}{|o_i|\cdot|o_{l:i}|}\left[\sum_{t=|o_l^i|}^{|o_i|}\min\left(r_{i,t}^{>l}(\theta),\text{clip}\left(r_{i,t}^{>l}(\theta),1\pm\epsilon\right)\right)\hat{A}_{i,t}\right]-\beta D_{\text{KL}}(\pi_\theta\|\pi_{\text{ref}})\\[2mm]
&= \frac{1}{G}\sum_{i=1}^{G}\frac{|o_l|}{|o_i|}\left[\frac{1}{|o_l|}\sum_{t=1}^{|o_l^i|}\min\left(r_{i,t}(\theta)^{<l},\text{clip}\left(r_{i,t}^{\leq l}(\theta),1\pm\epsilon\right)\right)\hat{A}_{i,t}\right.\\
&\qquad\qquad\left.-\frac{1}{|o_{l:i}|}\sum_{t=|o_l^i|}^{|o_i|}\min\left(r_{i,t}^{>l}(\theta),\text{clip}\left(r_{i,t}^{>l}(\theta),1\pm\epsilon\right)\right)\hat{A}_{i,t}\right]\\
&\qquad+\frac{1}{G}\sum_{i=1}^{G}\frac{1}{|o_{l:i}|}\left[\sum_{t=|o_l^i|}^{|o_i|}\min\left(r_{i,t}^{>l}(\theta),\text{clip}\left(r_{i,t}^{>l}(\theta),1\pm\epsilon\right)\right)\hat{A}_{i,t}\right]-\beta D_{\text{KL}}(\pi_\theta\|\pi_{\text{ref}})
\end{aligned}
\tag{14}
$$

We make the following definitions:

$$
J_i^{<l} = \frac{1}{|o_l|}\sum_{t=1}^{|o_l^i|}\min\left(r_{i,t}(\theta)^{<l},\text{clip}\left(r_{i,t}^{\leq l}(\theta),1\pm\epsilon\right)\right)\hat{A}_{i,t},
\tag{15}
$$

$$J_i^{>l} = \frac{1}{|o_{l:i}|} \sum_{t=|o_t^i|}^{|o_i|} \min\left( r_{i,t}^{>l}(\theta), \text{clip}\left( r_{i,t}^{>l}(\theta), 1 \pm \epsilon \right) \right) \hat{A}_{i,t}, \tag{16}$$

$$J_{\text{GRPO}}^{>l} = \frac{1}{G} \sum_{i=1}^{G} \frac{1}{|o_{l:i}|} \left[ \sum_{t=|o_t^i|}^{|o_i|} \min\left( r_{i,t}^{>l}(\theta), \text{clip}\left( r_{i,t}^{>l}(\theta), 1 \pm \epsilon \right) \right) \hat{A}_{i,t} \right], \tag{17}$$

where $J_i^{<l}$ represents the optimization objective of the shared tokens part of the $i$-th chain, $J_i^{>l}$ represents the optimization objective after partial rollout of the $i$-th chain, and $J_{\text{GRPO}}^{>l}$ represents the optimization objective of directly performing the classical GRPO sampling operation starting from the $l$-th position. Then, the original optimization objective $J_{\text{GRPO}}(\theta)$ can be expressed as:

$$J_{\text{GRPO}}(\theta) = \underbrace{\frac{1}{G} \sum_{i=1}^{G} \frac{|o_l|}{|o_i|} [J_i^{<l} - J_i^{>l}]}_{\text{Shared Token Advantage}} + J_{\text{GRPO}}^{>l} - \beta D_{\text{KL}}(\pi_\theta \parallel \pi_{\text{ref}})$$

$$= \underbrace{\frac{1}{G} \sum_{i=1}^{G} \frac{|o_l|}{|o_i|} [J_i^{<l}]}_{\text{Hard Advantage Estimation}} - \underbrace{\frac{1}{G} \sum_{i=1}^{G} \frac{|o_l|}{|o_i|} [J_i^{>l}]}_{\text{Regularization Term}} + J_{\text{GRPO}}^{>l} - \beta D_{\text{KL}}(\pi_\theta \parallel \pi_{\text{ref}}) \tag{18}$$

In this case, the GRPO optimization objective under Soft Advantage Estimation can be expressed as the sum of a standard GRPO objective from the partial rollout position, and the weighted difference between the objectives before and after that point. The weight of each difference is closely related to the length of its corresponding reasoning chain.

It is worth noting that, through our theoretical derivation and decomposition of the objective, we find that under the soft setting, the share token advantage consists of two components: **(1)** the first term aligns with the Hard Advantage Estimation; **(2)** the second term corresponds to the portion after the partial rollout position $l$, which is related to the individual token advantage. We interpret this second component as a regularization term. As a result, the estimated advantages under **the hard and soft settings are approximately equivalent, with the key distinction being the presence of this regularization term.** This leads us to hypothesize that **the improved stability of the Soft Advantage Estimation may be attributed to this regularization effect.**

### F.3 THEORETICAL PROOF OF GPG THEOREM

#### F.3.1 TRANSFORMER-BASED POLICY

The Transformer-based policy $\pi_\theta(a_t|s_t)$, by applying the chain rules, we have the following:

$$\begin{aligned}
&\pi_\theta(OT_1 \mid IT_1, IT_2, ..., IT_{|input|}) \times \\
&\pi_\theta(OT_2 \mid IT_1, IT_2, ..., IT_{|input|}, OT_1) \times \\
&\pi_\theta(OT_3 \mid IT_1, IT_2, ..., IT_{|input|}, OT_1, OT_2) \times \\
&... \\
&\pi_\theta(OT_{|output|} \mid IT_1, ..., IT_{|input|}, OT_1, ..., OT_{|output|-1}) \\
=&\pi_\theta(OT_1, OT_2, ..., OT_{|output|} \mid IT_1, IT_2, ..., IT_{|input|}) \\
=&\pi_\theta(MA \mid MS_1)
\end{aligned} \tag{19}$$

where $IT_i$ and $OT_i$ are input tokens and output tokens, respectively; $MS_1 \triangleq < IT_1, IT_2, ..., IT_{|input|} >$ and $MA \triangleq < OT_1, OT_2, ..., OT_{|output|} >$ can be token as the **macro state** and the **macro action**, respectively.

In a more general form, we can split the complete output $OT_1, OT_2, ..., OT_{|output|}$ into $K$ segments, and get the generalized macro states and macro actions, i.e., $MS_i \triangleq < MS_{i-1}, MA_{i-1} >$ and

$MA_i \triangleq < OT_m, OT_{m+1}, ..., OT_{m+n} >$. In this case, we have the following:

$$
\begin{aligned}
&\pi_\theta(MA \mid MS_1) \\
=&\pi_\theta(MA_1 \mid MS_1)\times \\
&\pi_\theta(MA_2 \mid MS_1, MA_1)\times \\
&... \\
&\pi_\theta(MA_K \mid MS_1, MA_1, MA_2, ...,, MA_{K-1}) \\
=&\pi_\theta(MA_1 \mid MS_1)\times \\
&\pi_\theta(MA_2 \mid MS_2)\times \\
&... \\
&\pi_\theta(MA_K \mid MS_K) \\
=&\prod_{T=1}^{K} \pi_\theta(MA_T \mid MS_T)
\end{aligned} \tag{20}
$$

where $T$ represents the **macro timestep**.

### F.3.2 DERIVATION OF THE GPG THEOREM

Given the macro states and macro actions defined above, we can get the Generalized Policy Gradient Theorem (for Transformer-based policies):

$$
\nabla_\theta J(\theta) = \mathbb{E}_{\tau \sim \pi_\theta} \{ \sum_{T=1}^{K} [\nabla_\theta \log \pi_\theta(MA_T | MS_T)\Phi_T] \} \tag{21}
$$

**A key advantage of the GPG Theorem is that it allows macro-action segmentation of arbitrary length**. This flexibility makes the theorem highly practical: for instance, enabling trajectory splitting based on special tokens.

The proof is as follows:

$$
\nabla_\theta J(\theta) \tag{22}
$$
$$
=\nabla_\theta \mathbb{E}_{\tau \sim \pi_\theta}[R(\tau)] \tag{23}
$$
$$
=\nabla_\theta \sum_\tau P(\tau;\theta)R(\tau) \tag{24}
$$
$$
=\sum_\tau \nabla_\theta P(\tau;\theta)R(\tau) \tag{25}
$$
$$
=\sum_\tau P(\tau;\theta)\frac{\nabla_\theta P(\tau;\theta)}{P(\tau;\theta)}R(\tau) \tag{26}
$$
$$
=\sum_\tau P(\tau;\theta)\nabla_\theta \log P(\tau;\theta)R(\tau) \tag{27}
$$
$$
=\sum_\tau P(\tau;\theta)\nabla_\theta[\log \mu(s_1)\prod_{t=1}^{H}\pi_\theta(a_t|s_t)P(s_{t+1}|s_t,a_t)]R(\tau) \tag{28}
$$
$$
=\sum_\tau P(\tau;\theta)\nabla_\theta[\log \prod_{t=1}^{H}\pi_\theta(a_t|s_t)P(s_{t+1}|s_t,a_t)]R(\tau) \tag{29}
$$
$$
=\sum_\tau P(\tau;\theta)\nabla_\theta[\log \prod_{t=1}^{H}\pi_\theta(a_t|s_t)]R(\tau) \tag{30}
$$
$$
=\sum_\tau P(\tau;\theta)\nabla_\theta[\log \prod_{T=1}^{K}\pi_\theta(MA_T|MS_T)]R(\tau) \tag{31}
$$
$$
=\sum_\tau P(\tau;\theta)[\sum_{T=1}^{K}\nabla_\theta \log \pi_\theta(MA_T|MS_T)]R(\tau) \tag{32}
$$

$$= \sum_\tau P(\tau; \theta) \left[ \sum_{T=1}^{K} \nabla_\theta \log \pi_\theta(MA_T|MS_T) R(\tau) \right] \tag{33}$$

$$= \mathbb{E}_{\tau \sim \pi_\theta} \left\{ \sum_{T=1}^{K} [\nabla_\theta \log \pi_\theta(MA_T|MS_T) R(\tau)] \right\} \tag{34}$$

$$= \mathbb{E}_{\tau \sim \pi_\theta} \left\{ \sum_{T=1}^{K} [\nabla_\theta \log \pi_\theta(MA_T|MS_T) \Phi_T] \right\} \tag{35}$$

Key steps in the proof are presented below:

1. From Equation (29) to Equation (30), this is because $s_{t+1} = [s_t, a_t]$ for Transformer-based policy, so we have $P(s_{t+1}|s_t, a_t) = 1$.

2. From Equation (30) to Equation (31), this is because $s_{t+1} = [s_t, a_t]$ for Transformer-based policy, so we have the following derivation:

$$\begin{aligned}
&\prod_{t=1}^{H} \pi_\theta(a_t|s_t) \\
=&\pi_\theta(a_1|s_1) \times \pi_\theta(a_2|s_2) \times ... \times \pi_\theta(a_H|s_H) \\
=&\pi_\theta(a_1|s_1) \times \pi_\theta(a_2|s_1, a_1) \times ... \times \pi_\theta(a_H|s_0, a_0, a_1, ..., a_{H-1}) \\
=&\pi_\theta(a_1, a_2, ..., a_H|s_1) \\
=&\pi_\theta(MA \mid MS_1) \\
=&\pi_\theta(MA_1 \mid MS_1) \times \\
&\pi_\theta(MA_2 \mid MS_1, MA_1) \times \\
&... \\
&\pi_\theta(MA_K \mid MS_1, MA_1, MA_2, ..., , MA_{K-1}) \\
=&\prod_{T=1}^{K} \pi_\theta(MA_T \mid MS_T)
\end{aligned} \tag{36}$$

3. From Equation (34) to Equation (35), this is similar to the transition from Policy Gradient Theorem to the general form of Policy Gradient Theorem (i.e., from traditional advantage function to GAE).

## G USE OF LLMS

In this paper, the LLM is used solely for correcting grammatical and lexical errors.

# H  THE ALGORITHM WORKFLOW OF ARPO

In this section, we provide a detailed flowchart of the ARPO algorithm in diagram 1.

---

**Algorithm 1** Agentic Reinforce Policy Optimization

---

**Input** initial policy model $\pi_{\theta_{\text{init}}}$; reward models $r_\phi$; task prompts $\mathcal{D}$; hyperparameters $\epsilon, \alpha, \beta, \mu, \tau,$ $M, N, Z, k$

1: policy model $\pi_\theta \leftarrow \pi_{\theta_{\text{init}}}$
2: **for** iteration = 1, ..., I **do**
3:     reference model $\pi_{ref} \leftarrow \pi_\theta$
4:     **for** step = 1, ..., S **do**
5:         Sample a batch $\mathcal{D}_b$ from $\mathcal{D}$
6:         Update the old policy model $\pi_{\theta_{old}} \leftarrow \pi_\theta$
7:         Sample $N$ reasoning 1-step paths $\{y_i\}_{i=1}^N \sim \pi_{\theta_{old}}(\cdot \mid q)$ for each question $q \in \mathcal{D}_b$
8:         Compute initial entropy $H_{i,initial}$ of the first $k$ tokens in each outputs
9:         let all rollouts $\{rollout_i\} \leftarrow \{y_i\}_{i=1}^N$
10:        tool-call step $t \leftarrow 1$
11:        **while** Any unfinished $y_i$ **do**
12:            Parse unfinished $y_i$, execute tools and obtain results $d_i = T(y_i)$
13:            Insert $d$ into rollout $y_i \leftarrow y_i + d_i$
14:            Generate $k$ additional tokens based on inserted $y_i$ to compute step-level entropy $H_{i,t}$
15:            Compute normalized change in entropy $\Delta H_{i,t} = Normalize(H_{i,t} - H_{i,inintial})$
        for each $y_i$
16:            Compute partial sampling probability $P_{i,t} = \alpha + \beta \cdot \Delta H_{i,t}$
17:            **if** $P_{i,t} > \tau$ **then**
18:                Branch out $Z$ additional rollouts $\{y_i\}_i^Z$ and add them to $\{rollout_i\}$
19:            **if** $|y_i| = M$ **then**
20:                Sample $\{y_i\}$ until $y_i$ produce the final answer
21:            **else**
22:                Sample $\{y_i\}$ to produce new reasoning step
23:                $t \leftarrow t + 1$
24:        **if** $|y_i| < M$ **then**
25:            Sample $M - |y_i|$ additional independent rollouts and add them to $\{rollout_i\}$
26:            Compute rewards $\{r_i\}_{i=1}^M$ for each sampled rollouts $y_i$ by running $r_\phi$
27:            Compute $\hat{A}_{i,t}$ for the $t$-th token of $o_i$ through group relative advantage estimation.
28:        **for** GRPO iteration = 1, ..., $\mu$ **do**
29:            Update the policy model $\pi_\theta$ by maximizing the GRPO objective (Equation (3))
30:     Update $r_\phi$ through continuous training using a replay mechanism.

**Output** $\pi_\theta$

---

Table 6: An example from ARPO on GAIA dataset, with special symbols used in think content, search queries, Python codes, returned results and final answer highlighted with `purple box`, `green box`, `blue box` and `red box`, respectively.

---

**Example #2 from GAIA Dataset**

**Question:**

In terms of geographical distance between capital cities, which 2 countries are the furthest from each other within the ASEAN bloc according to wikipedia? Answer using a comma separated list, ordering the countries by alphabetical order.

**Labeled Answer:**

Indonesia, Myanmar

---

**Model Output by ARPO(14B)**

`<think>` Okay, so I need to figure out which two ASEAN member countries have their capital cities that are the furthest apart geographically. And then present them in alphabetical order, comma-separated. Let me start by recalling what ASEAN stands for—it's the Association of Southeast Asian Nations. The current members include Brunei, Cambodia, Indonesia, Laos, Malaysia, Myanmar (Burma), Philippines, Singapore, Thailand, Vietnam, and maybe others like East Timor or Papua New Guinea? Wait, no, actually I think it's those ten countries. Let me confirm that first. (***More reasoning contentare omitted here***) First, let me gather the list of capitals and their coordinates. I should search for ÄSEAN member countries and their capitals̈to ensure accuracy. Also, maybe check each country's Wikipedia page for their capital's coordinates. Alternatively, look up a list that includes both names and coordinates. Let me try a search query first. `</think>`

`<search>` List of ASEAN countries with capitals and geographic coordinates `</search>`

`<result>`

The searched web pages provide the list of ASEAN member countries along with their capitals but do not include geographic coordinates for the capitals. For example: - **Brunei**: Capital is **Bandar Seri Begawan**. - **Cambodia**: Capital is **Phnom Penh**. - **Indonesia**: Capital is **Jakarta**. - **Laos**: Capital is **Vientiane**. - **Malaysia**: Capital is **Kuala Lumpur**. - **Myanmar**: Capital is **Naypyidaw**. - **Philippines**: Capital is **Manila**. - **Singapore**: Capital is **Singapore**. - **Thailand**: Capital is **Bangkok**. - **Vietnam**: Capital is **Hanoi**. To proceed with calculating distances between capitals, geographic coordinates (latitude and longitude) for each capital would need to be obtained from an external source.

`</result>`

***Two search steps omitted here***

`<think>` So yes, exactly 10 countries. Therefore, the number of pairs is C(10,2)=45. Computing all 45 isn't too bad for a script.(***More reasoning contentare omitted here***) Given all these calculations, I'm confident that the furthest pair is Indonesia and Myanmar. Their capitals are Jakarta and Naypyidaw respectively. Sorting alphabetically, Indonesia comes before Myanmar. Hence the answer should be Indonesia,Myanmar. To determine the two ASEAN member countries with the furthest apart capitals, I first identified the capitals and their geographic coordinates from reliable sources. Using the Haversine formula, I calculated the distances between all pairs of capitals. After performing these calculations, the maximum distance was found between **Jakarta (Indonesia)** and **Naypyidaw (Myanmar)**. When sorted alphabetically, the countries are **Indonesia** and **Myanmar**. `</think>`

`<answer>`

*Indonesia, Myanmar*

`</answer>`

---

# I CASE STUDY

Table 7: An example from ARPO on HLE dataset, with special symbols used in think content, search queries, Python codes, returned results and final answer highlighted with `purple box`, `green box`, `blue box` and `red box`, respectively.

---

**Example #2 from HLE Dataset**

---

**Question:**
How long was the Second Great War in StarCraft Lore (in years, rounded up)
**Labeled Answer:**
1

---

**Model Output by ARPO(14B)**

`<think>` Okay, so I need to figure out how long the Second Great War lasted in StarCraft lore, right? Let me start by recalling what I know about StarCraft's timeline. **(\*\*\*More reasoning contentare omitted here\*\*\*)** So if it starts in 2500 and ends in 2506, that's seven years. But maybe the core conflict is counted until 2505, making it five years. Alternatively, some sources might consider the entire period from the first signs of the Zerg resurgence up to the final battles as spanning six years. `</think>`

`<search>` starcraft second great war duration years `</search>`

`<result>`
The Second Great War in *StarCraft* is generally considered to have occurred from \*\*2504 to 2505\*\*, spanning \*\*one year\*\*. This timeline aligns with the events of *StarCraft II: Wings of Liberty* (2504) and *Heart of the Swarm* (2505). However, some sources, such as NamuWiki, specify that the war lasted for half a year until April 2505, suggesting the \*\*core conflict\*\* may have been shorter (six months), though the broader war period is recorded as one year. The war followed the end of the Brood War (2500) and preceded the *Legacy of the Void* campaign.
`</result>`

**\*\*\*Two search steps omitted here\*\*\***

`<think>` Okay, so the official StarCraft Wiki states it's one year. Even if parts of it were shorter, the overall duration is considered a full year. Therefore, rounding up wouldn't change it since it's already a whole number. But just to be thorough, maybe there's a different interpretation. Let me see if any other sources say otherwise.Alternatively, perhaps the Second Great War includes the events leading up to the main games. For example, if the war started in late 2503 and ended in mid-2505, that would be almost two years. But according to the wiki, it's 2504-2505. Given that the user specified rounded up, even if it were, say, 1.5 years, we'd round to 2. But based on the search results, especially the official wiki, it's one year. So the answer should be 1 year. Wait, but the first search had conflicting info. Let me recheck the exact wording from the wiki.Assuming the official timeline clearly states 2504–2505 as the Second Great War, that's exactly one year. Hence, the duration is one year. Rounded up, it remains one. Therefore, the answer is 1 year. But wait, hold on! Another possibility: the Second Great War might refer to the conflict in the original StarCraft games versus the sequel. No, the Second Great War is definitely part of the StarCraft II storyline. Let me confirm once more.Yes, the Second Great War is indeed the central conflict of StarCraft II, occurring between 2504 and 2505, totaling one year. Some sources might mention the core battle lasting six months, but the entire war period is a year. `</think>`

`<answer>`
Since the question asks for the duration in years, rounded up, and it's exactly one year, no rounding is needed.
1
`</answer>`

Table 8: An example from ARPO on AIME24 dataset, with special symbols used in think content, search queries, Python codes, returned results and final answer highlighted with `purple box`, `green box`, `blue box` and `red box`, respectively.

### Example #1 from AIME24 Dataset

**Question:**
Let $p$ be the least prime number for which there exists a positive integer $n$ such that $n^4 + 1$ is divisible by $p^2$. Find the least positive integer $m$ such that $m^4 + 1$ is divisible by $p^2$.

**Labeled Answer:**
110

### Model Output by ARPO(8B)

`<think>` To solve this problem, we need to find the smallest prime $p$ such that there exists a positive integer $n$ for which $n^4 + 1$ is divisible by $p^2$. Then, we need to find the smallest positive integer $m$ such that $m^4 + 1$ is divisible by $p^2$. We will start by checking small prime numbers one by one to find the smallest $p$ that satisfies the condition. We'll use Python to automate this process. First, let's write a function to check if $n^4 + 1$ is divisible by $p^2$. We'll iterate over the first few values of $n$ to find such an $n$ for each prime $p$. We'll start with the smallest prime number and move upwards until we find the required $p$. Here's the plan: 1. Iterate over the first few prime numbers. 2. For each prime $p$, check if there exists an $n$ such that $n^4 + 1 \equiv 0 \pmod{p^2}$. 3. Once we find the smallest prime $p$ that satisfies the condition, we'll find the smallest $m$ such that $m^4 + 1 \equiv 0 \pmod{p^2}$. Let's implement this in Python. `</think>`

`<python>`

```python
from sympy import isprime, primerange
# Function to check if there exists an n such that n^4 + 1 is divisible
    by p^2
def find_n_for_p(p):
    for n in range(1, p**2 + 1):
        if (n**4 + 1) % (p**2) == 0:
            return n
        return None
# Iterate over the first few prime numbers to find the smallest p
primes = list(primerange(2, 100))
# We'll start checking from the first few primes
for p in primes:
    n = find_n_for_p(p)
    if n is not None:
        # Now we need to find the smallest m such that m^4 + 1 is
            divisible by p^2
        for m in range(1, p**2 + 1):
            if (m**4 + 1) % (p**2) == 0:
                result = m
                break
            else:
                continue
            break
print((p, result))
```

`</python>`

`<think>` The smallest prime $p$ for which there exists a positive integer $n$ such that $n^4 + 1$ is divisible by $p^2$ is 17. The least positive integer $m$ such that $m^4 + 1$ is divisible by $17^2$ is 110. Thus, the final answer is: `</think>`

`<answer>`

110

`</answer>`

Table 9: An example from ARPO on HLE dataset, with special symbols used in think content, search queries, Python codes, returned results and final answer highlighted with `purple box`, `green box`, `blue box` and `red box`, respectively.

---

**Example #1 from HotpotQA Dataset**

**Question:**
Which playwright lived a longer life, Edward Albee or J. M. Barrie?
**Labeled Answer:**
Edward Franklin Albee III

---

**Model Output by ARPO(8B)**

`<think>` I need to determine which of two playwrights, Edward Albee and J. M. Barrie, lived a longer life. To do this, I will: 1. Find the birth date of Edward Albee. 2. Find the death date of Edward Albee. 3. Calculate how long Edward Albee lived. 4. Find the birth date of J. M. Barrie. 5. Find the death date of J. M. Barrie. 6. Calculate how long J. M. Barrie lived. 7. Compare the lifespans. I will use web search for steps 1-5. `</think>`
`<search>` Edward Albee birth death dates `</search>`
`<result>`

Page 1: Sep 16, 2016 · Edward Albee was born Edward Harvey in Washington, D.C. At the age of two weeks, he was adopted by Mr. and Mrs. Reed Albee of Larchmont, New York, and renamed Edward Franklin Albee III. From an early age, …

Page 2: Explore Edward Albee's biography, personal life, family and cause of death. Discover the real story, facts, and details of Edward Albee.

Page 3: Sep 17, 2016 · Edward Albee was a Pulitzer Prize winning American playwright. His popular works include The Zoo Story, and Who's Afraid of Virginia Wolf? Albee was born Edward Harvey on March 12, 1928 in Washington, D.C. (***More search results omitted here***)
`</result>`
`<think>` From the search results: - Edward Albee was born on March 12, 1928. - Edward Albee died on September 16, 2016. Now I need to find J. M. Barrie's birth and death dates. `</think>`
***One search step omitted here***
`<think>` From the search results:- J. M. Barrie was born on May 9, 1860. - J. M. Barrie died on June 19, 1937. Now I have both sets of dates:
Edward Albee: - Born: March 12, 1928 - Died: September 16, 2016
J. M. Barrie: - Born: May 9, 1860 - Died: June 19, 1937
Next, I will calculate how long each playwright lived. For Edward Albee: 2016 - 1928 = 88 years. For J. M. Barrie: 1937 - 1860 = 77 years. So, Edward Albee lived 88 years and J. M. Barrie lived 77 years. Therefore, Edward Albee lived a longer life. `</think>`
`<python>`

```python
# Calculate lifespan for Edward Albee
albee_birth_year = 1928
albee_death_year = 2016
albee_lifespan = albee_death_year - albee_birth_year
print(f"Edward Albee lived for {albee_lifespan} years.")
# Calculate lifespan for J. M. Barrie
barrie_birth_year = 1860
barrie_death_year = 1937
barrie_lifespan = barrie_death_year - barrie_birth_year
print(f"J. M. Barrie lived for {barrie_lifespan} years.")
# Compare lifespans
if albee_lifespan > barrie_lifespan:
    print("Edward Albee lived a longer life.")
else:
    print("J. M. Barrie lived a longer life.")
```

`</python>`
`<result>`
Edward Albee lived for 88 years. J. M. Barrie lived for 77 years. Edward Albee lived a longer life.
`</result>`
`<answer>`
Edward Albee
`</answer>`

