# OpenReview forum: "Agentic Reinforced Policy Optimization"
_ICLR.cc/2026/Conference — ICLR 2026 Poster_

### Official Review · Reviewer_BZ6R · 2025-10-27

**Soundness:** 4
**Presentation:** 3
**Contribution:** 2
**Rating:** 4
**Confidence:** 3

**Summary:**

This paper presents a sampling strategy, which generates additional trajectories by forking the trajectory at timesteps where the tool call probabilities indicate high entropy, and use this strategy as part of the rollout stage for multi-turn agentic RL training for LLMs. The presented method (ARPO) provides consistent improvement in a variety of tasks (math, coding, deep search) and highlights the importance of further research in exploration within the area of RLVR for LLMs.

**Strengths:**

- Paper is well motivated and the observation of high entropy tool calling steps is clearly presented.
- Consistent improvements in evaluated domains (math, coding, deep search).
- Mathematical formulation is grounded and highlights the efficacy of their sampling approach.

**Weaknesses:**

- Limited Novelty: The main contribution seems to be the additional sampling introduced by branching from high entropy actions. The learning objective is simply multi-turn GRPO given rollouts generated by their sampling mechanism.
- Effects of branching sampling ($P_{t}$) is unclear. There are no ablations documenting the effects of relying on the base probability ($\alpha$) versus relying on the entropy differential ($\beta$) or the branching cutoff ($\tau$).
- Unclear if the effectiveness of high-entropy branching is unique to tool-calling steps or broadly applicable (e.g. to text-based reasoning steps)

**Questions:**

1. Could the authors quantify the effects of varying $\alpha$, $\beta$, and $\tau$ during sampling? The entropy figure in the Appendix indicates that if entropy is too high performance degrades. Thus, understanding the relationship between base model probs and entropy during sampling seems crucial.
2. Are there other actions, besides high entropy tool calls, where branching is effective? The authors seem to only branch at high-entropy tool calling steps, but I wonder if this method is generally applicable to reasoning steps too.

---

> ### Author Response · Authors · 2025-11-24
> **Response 1/2**
>
> We sincerely thank the reviewer for the thorough and constructive feedback. The reviewer's careful examination of our work have been invaluable in helping us clarify and strengthen our contributions. Below, we address each concern systematically and comprehensively.
>
> ---
>
> ### W1: Main Contribution of ARPO Limited to GRPO-based Extra Branching Sampling
>
> We thank the reviewer for raising this important concern regarding the novelty of our work. We would like to clarify that **ARPO is not simply "adding extra sampling on top of GRPO."** We articulate ARPO's core contributions from three distinct perspectives:
>
> **1. Motivation Level: Establishing Algorithmic Mechanisms from Inherent Characteristics of Agentic Reasoning**
>
> Existing agentic RL methods typically rely on reward adjustments, additional objectives, or heuristic sampling rules. They lack investigation grounded in the inherent behavioral patterns of LLM-based agents during multi-turn tool-calling reasoning processes (multi-turn interactions, entropy dynamics, long context, etc.). In contrast, through preliminary exploration, ARPO pioneers an approach that starts from the inherent characteristics of multi-turn agentic reasoning with tool-environment interactions. We discovered that external tool results trigger model uncertainty and further revealed a consistent entropy increase phenomenon before and after tool calls. Based on this inherent pattern, we naturally introduced **entropy-based adaptive rollout**—a simple yet effective exploration mechanism specifically designed for multi-turn agentic reasoning. Furthermore, ARPO employs Advantage Attribution Estimation to assist the LLM in better internalizing advantage differences in stepwise tool-use behaviors. Therefore, our method is deeply aligned with its motivation, providing a novel direction for understanding and leveraging the "exploration and exploitation" trade-off in agentic behavior, rather than relying on heuristic modifications to reward functions—an aspect not addressed by previous work.
>
> **2. Advantage Estimation Level: Mathematically Distinct Advantage Estimation from GRPO**
>
> ARPO does not simply reuse GRPO's objective. We first introduce and compare two settings: hard and soft advantage estimation, and reveal stability analyses of both through experimental results. To further elucidate the distinction between these two estimation methods, as demonstrated in the theoretical analysis in `Appendix F.2 "Theoretical Analysis of Soft Advantage Estimation,"` we theoretically show that the key advantage of soft advantage estimation over hard advantage lies in a regularization term related to response length, which significantly impacts training stability. Additionally, we would like to clarify that while soft advantage estimation uses GRPO's original formula, the partial adaptive rollout mechanism generates trajectories with shared semantic prefixes. This naturally causes the importance sampling (IS) ratio to differentiate between shared tokens and branched tokens. Through this mechanism, which is incorporated into Equation (3), ARPO naturally distinguishes between shared trajectories and branched trajectories, thereby forming fundamentally different gradient contributions. Therefore, ARPO's optimization objective is mathematically distinct from standard GRPO.
>
> **3. Theoretical and Empirical Levels: GPG Theorem and Superior Empirical Results**
>
> Beyond empirically proposing ARPO, we provide strong theoretical foundations. Theoretically, in `Section 3.3 "Theoretical Foundation"` and `Appendix F.3 "Theoretical Proof of GPG Theorem,"` we derive the generalized policy gradient (GPG) form of ARPO and prove the algorithmic convergence of this framework for Transformer-structured models. We further theoretically analyze the distinction between soft and hard advantage settings in `Appendix F.2 "Theoretical Analysis of Soft Advantage Estimation."` Empirically, under conditions where tool-calling budgets are reduced by approximately 40\%–50\%, ARPO significantly outperforms both GRPO[1] and REINFORCE++[2] across 10 reasoning tasks (see Table 1 and Figure 7). Therefore, we believe that ARPO provides a pioneering and theoretically grounded approach to the "exploration and exploitation" trade-off in agent reinforcement learning, with contributions that extend well beyond simple sampling modifications.
>
> ---
>
> #### W2 & Q1: Lack of Hyperparameter Selection Analysis
>
> We thank the reviewer for this valuable feedback. While we have analyzed the impact of the entropy differential weight ($\beta$), initial sampling size ($N$), and global rollout size ($M$) in `Appendix A.2 "Scaling Analysis of ARPO,"` we recognize that a more systematic examination of the base probability ($\alpha$) and branching cutoff ($\tau$) would strengthen our analysis.
>
> Threfore, we have conducted targeted ablation studies focusing on the effects of $\alpha$ and $\tau$. Below, we present our experimental setup and findings:

---

> ### Author Response · Authors · 2025-11-24
> **Response 2/2**
>
> **Experimental Setup**: We use Qwen2.5-7B[3] as the backbone model and evaluate on three core tasks: GAIA[4] (deep search), AIME24 (mathematical reasoning), and HotpotQA[5] (multi-hop knowledge reasoning). We fix the global rollout size $M=16$, initial sampling size $N=8$, and entropy differential weight $\beta=0.2$. We then independently vary the base probability $\alpha$ and the branching cutoff $\tau$ using a controlled variable approach, with Pass@1 accuracy as the primary metric.
>
> | Parameter Setting | GAIA  | AIME24| HotpotQA | Average  |
> |---|:---:|:---:|:----:|:----:|
> | $\tau=0.5$ (standard) | 43.7% | 30.0% | 58.8% | **44.2%** |
> | $\tau=0.3$ (lenient) | 41.9% | 26.6% | 57.6% | 42.0% |
> | $\tau=0.7$ (strict) | 39.4% | 26.6% | 55.3% | 40.4% |
> | $\alpha=0.5$ (standard) | 43.7% | 30.0% | 58.8% | **44.2%** |
> | $\alpha=0.0$ (no base probability) | 38.5% | 23.3% | 55.2% | 39.0% |
> | $\alpha=1.0$ (always branch) | 35.6% | 20.0% | 52.9% | 36.1% |
>
> The ablation studies reveal that $\alpha$ provides fundamental exploration coverage: when $\alpha=0.0$ (entropy-only branching), performance drops by **5.2 percentage points**, demonstrating that a baseline branching probability is essential even when entropy changes are minimal. For $\tau$, which controls the branching intensity-efficiency trade-off, $\tau=0.3$ causes moderate degradation (**2.2 percentage points**), while $\tau=0.7$ results in **3.8 percentage point** drop by eliminating valuable branches. These results confirm that $\tau=0.5$ achieves optimal balance between branching coverage and computational efficiency.
>
> We have incorporated these detailed ablation studies into `Appendix A.3 "Ablations on Branching Sampling Parameter"` to improve the interpretability and reproducibility of our method.
>
> ---
>
> ### W3 & Q2: Uncertain Applicability of High-Entropy Branching Beyond Tool-Call Steps
>
> We thank the reviewer for raising these critical questions. We would like to clarify that **ARPO fundamentally incorporates high-entropy signals of policy uncertainty rather than depending on the specific form of "tool-call actions."** To validate that the branching mechanism is theoretically applicable to general text reasoning steps, we conduct additional experiments using a more general chunk-level rollout branching design of ARPO.
>
> **Chunk-level Branch Design**: Given a global rollout size $M$, during the RL process the model generates $N$ initial sampling chains. We divide each reasoning chain into semantic chunks (each chunk has a length of approximately 400 tokens). For each chunk, we compute the average policy entropy based on its output distribution and apply ARPO's branching formula (Equation 2). Specifically, when a chunk's token entropy is significantly higher than the baseline problem entropy, the model branches after that chunk to focus on exploring the reasoning segments where the model exhibits the highest uncertainty.
>
> | Method | AIME24 | MATH[6] | MATH500[7] | WebWalker[8] | Bamboogle[9] | HotpotQA[5] | Avg. |
> |----|:---:|:--:|:---:|:---:|:---:|:----:|:--:|
> | Qwen2.5-7B[3] + GRPO[1] | 23.3% | 87.8% | 78.0% | 22.0% | 68.4% | 59.0% | 56.4% |
> | Qwen2.5-7B[3] + ARPO (chunk-level) | **26.6%** | **88.8%** | **80.4%** | **26.0%** | **71.5%** | **60.7%** | **59.0%** |
> | **Improvement** | +3.3% | +1.0% | +2.4% | +4.0% | +3.1% | +1.7% | +2.6% |
>
> **Results and Analysis**: The experimental results are presented in the following table. ARPO, under the chunk-level setting, consistently improves accuracy across mathematical and knowledge reasoning tasks, achieving an average improvement of **+2.6%** over GRPO[1].This finding indicates that the branching mechanism naturally generalizes beyond tool-calling steps and is fundamentally driven by entropy-based uncertainty detection rather than domain-specific heuristics.
>
> We provide detailed results and additional analysis in `Appendix A.7 "Generalization to Text-only Reasoning: Chunk-Level Branching Analysis"` to demonstrate the general applicability of ARPO's entropy-based branching mechanism.
>
> ---
>
> ### Response Summary
>
> To conclude, our comprehensive rebuttal addresses all reviewer concerns through detailed explanations, experiments, and theoretical clarifications. Specifically, we have:
>
> **(1)** elucidated ARPO's distinct contributions at three levels—motivational grounding, mathematically distinct advantage estimation from GRPO[1], and theoretical foundations via the GPG theorem;
>
> **(2)** conducted extensive hyperparameter ablation studies on $\alpha$ and $\tau$, revealing their roles and optimal configurations;
>
> **(3)** validated ARPO's generalizability through chunk-level experiments, confirming extension beyond tool-calling to text reasoning.
>
> We believe these responses demonstrate ARPO's significance, effectiveness, and methodological completeness. We are deeply grateful to the reviewer for the thorough review and insightful questions, which have been invaluable in clarifying and strengthening our contributions.

---

> > ### Author Response · Authors · 2025-11-24
> > **Reference**
> >
> > [1] Group Relative Policy Optimization
> >
> > [2] REINFORCE++: A Simple and Efficient Approach for Aligning Large Language Models
> >
> > [3] Qwen2.5 Technical Report
> >
> > [4] GAIA: a benchmark for General AI Assistants
> >
> > [5] HotpotQA: A Dataset for Diverse, Explainable Multi-hop Question Answering
> >
> > [6] Measuring Mathematical Problem Solving with the MATH Dataset
> >
> > [7] MATH500 Dataset (selected from MATH)
> >
> > [8] WebWalker: Benchmarking LLMs in web traversal
> >
> > [9] Measuring and narrowing the compositionality gap in language models

---

> ### Author Response · Authors · 2025-11-27
> **A Kind Reminder for Reading the Response**
>
> Dear Reviewer BZ6R,
>
> We have revised the paper and included additional entropy-related and hyper-parameter selection analyzes. In the response above, we have also tried to clarify our contributions over previous (concurrent) works. Since the rebuttal period is closing very soon, could you please check our response to see whether it mitigates your concerns? We would greatly appreciate that!
>
> Thank you, The authors of ARPO

---

### Official Review · Reviewer_tZJ2 · 2025-10-28

**Soundness:** 3
**Presentation:** 3
**Contribution:** 2
**Rating:** 4
**Confidence:** 4

**Summary:**

This paper introduces ARPO, a RL algorithm designed to improve LLM agents that use external tools in multi-turn interactions. The key novelty is an entropy-based adaptive rollout mechanism that dynamically performs additional sampling at high-entropy tool-call steps and an advantage attribution estimation method that distinguishes shared vs. branched trajectories. Experiments across 13 benchmarks (math, knowledge reasoning, and deep search tasks) show that ARPO improves accuracy and efficiency, reportedly achieving similar or better performance with half the tool-use budget compared to GRPO and other baselines.

**Strengths:**

- The evaluation covers a wide range of tasks (reasoning, knowledge retrieval, deep search), showing consistent improvements over multiple baselines (GRPO, DAPO, Reinforce++).
- ARPO achieves better or comparable results with roughly half the tool-call budget, which is practically valuable given the high cost of tool-based RL.
- The GPG theorem provides a solid conceptual framework connecting macro-step rollouts to standard policy gradients, offering a theoretical foundation.
- The paper is well-organized, with clear modular explanations of the rollout mechanism and advantage estimation, supported by informative figures.

**Weaknesses:**

- The paper’s central motivation that token entropy reliably increases after tool calls is not convincingly demonstrated. Figures 1 and 2 appear anecdotal, with unclear statistical support, and may rely on a single sample.
- While ARPO’s adaptive rollout is designed to mitigate high-entropy uncertainty, the paper does not show post-training token entropy patterns to demonstrate that the method actually reduces or stabilizes entropy. Without before-and-after comparisons, it’s unclear whether ARPO genuinely controls model uncertainty or simply improves performance indirectly.
- The experiments only report numerical performance gains, without showing any entropy-related observations or analyses during or after training. This creates a disconnect between the paper’s motivation entropy-guided exploration and its empirical validation, weakening the causal link between the proposed mechanism and the observed improvements.

**Questions:**

1. The paper’s motivation relies on the observation that token entropy tends to increase after tool-call steps, which is claimed to indicate model uncertainty. Could the authors provide quantitative statistics (e.g., average entropy changes, variance, and significance tests) across multiple datasets and trajectories to confirm that this pattern consistently holds?
2. Since ARPO is designed to mitigate high-entropy uncertainty, it would be important to show entropy evolution before and after training. Do the authors have measurements or visualizations demonstrating that entropy indeed decreases or stabilizes after ARPO training?
3. Did the authors observe any instability or divergence issues during training due to dynamic branching?
4. It would be helpful to include a PPO baseline or comparison, given that PPO is a standard RL method.

---

> ### Author Response · Authors · 2025-11-24
> **Response 1/2**
>
> We sincerely thank the reviewer for the thorough and constructive feedback. The reviewer's detailed observations and suggestions have been invaluable in identifying areas for improvement. Below, we address each concern systematically and comprehensively.
>
> ---
>
> ### W1 & Q1: Limited Statistical Evidence for Entropy Increase After Tool Calls
>
> We thank the reviewer for raising this important point. We fully acknowledge that the initial presentation of entropy observations in Figures 1 and 2 may have appeared anecdotal. We have therefore conducted comprehensive statistical analyses to quantitatively validate the entropy increase phenomenon across multiple datasets.
>
> **Statistical Analysis**: We quantitatively measured entropy changes across reasoning datasets (AIME24, AIME25, MATH[1], GSM8K[2]) and deep-search datasets (GAIA[3], HLE[4], WebWalker[5], XBench[6]). For each dataset, we computed the average entropy of the first 20 tokens following each tool call using the formula $\Delta H_t = \text{Normalize}(H_t - H_{\text{initial}})$, comparing it against the entropy of the initial question tokens. We then calculated the proportion of tool calls classified as "high-entropy calls" across all tool calls in each dataset.
>
> **Results and Analysis**: The statistical results are presented in Table below, our analyses are listed as follow:
>
> | Metric | GAIA | HLE | Webwalker | Xbench | AIME24 | AIME25 | Math | GSM8K |
> |----|:--:|:--:|:----:|:----:|:----:|:---:|:---:|:----:|
> | **High-Entropy Tool Calls (Avg.)** | 3.43 | 1.36 | 1.85 | 2.81 | 0.54 | 0.69 | 0.45 | 0.39 |
> | **Total Tool Calls (Avg.)** | 6.41 | 2.84 | 4.22 | 6.16 | 1.23 | 1.67 | 1.01 | 1.02 |
> | **High-Entropy Ratio** | **53.5%** | **47.9%** | **43.8%** | **45.6%** | **43.9%** | **41.3%** | **44.6%** | **38.2%** |
>
> Across all eight evaluated datasets, **over 38% of tool calls exhibit significant entropy increase**, with the ratio ranging from 38.2% (GSM8K[2]) to 53.5% (GAIA[3]). This demonstrates that the entropy elevation pattern is **stable and persistent** across different task domains and not attributable to individual outlier cases.
>
> We have incorporated these statistical analyses and detailed results into the `Appendix A.1 "Statistical Analysis of Tool-call Entropy Dynamic"` to provide quantitative evidence for supporting our motivations.
>
> ---
>
> ### W2, W3 & Q2: Missing Evidence of Entropy Reduction or Stabilization in ARPO Training. Please provide measurements or visualizations.
>
> We appreciate the reviewer's constructive feedback: **the core objective of ARPO is not to reduce or suppress high token entropy**. Instead, our method is designed to **identify and leverage the natural entropy increase that occurs after tool calls, as these high-entropy regions often signal richer potential solution spaces.** ARPO's dynamic branching mechanism performs additional sampling precisely at these high-entropy steps, thereby more thoroughly exploring diverse reasoning paths. By transforming high-uncertainty regions into structured exploration opportunities, ARPO enhances the coverage of the solution space during the rollout phase.
>
> To quantify the advantages of ARPO beyond mere performance improvements, we have designed two experiments:
>
> **1. Rollout Sampling Diversity Analysis**: To demonstrate that ARPO achieves better coverage of the rollout solution space compared to GRPO[7], we randomly sampled 640 problems from 10 rollout steps, collecting approximately 7.6k trajectories. Using BGEM3 as the semantic embedding model, we visualized the sampling distribution of rollouts through PCA dimensionality reduction[8] and DBSCAN clustering[9]. As shown in  `Section 4.3 "Rollout Sampling Diversity Analysis,"` ARPO's sampling trajectories form more distinct and clearer cluster centers (54 clusters vs. 48 for GRPO[7]), with greater intra-cluster compactness and larger inter-cluster separation. These findings indicate that **ARPO effectively exploits the transformation of high-entropy uncertainty into exploration opportunities, significantly improving rollout diversity and making the distribution of sampled paths more structured.**
>
> **2. Policy Entropy Dynamics Analysis**: To accurately demonstrate the effects of ARPO's entropy-guided policy optimization, we compared GRPO[7], CISPO[10], and ARPO by plotting reinforcement learning training curves across 10 reasoning tasks. As shown in `Section 4.3 "Effect of Policy Entropy on Performance,"` we present the entropy loss and corresponding validation accuracy changes throughout the entire training process for different RL algorithms. The results show that ARPO exhibits a more stable entropy loss while achieving stronger accuracy. Notably, both CISPO[10] and GRPO[7] exhibit larger entropy fluctuations, which do not translate into improved training effectiveness. In contrast, **ARPO maintains consistently higher and more stable entropy levels throughout training, which facilitates steady and reliable performance improvements.**

---

> ### Author Response · Authors · 2025-11-24
> **Response 2/2**
>
> ### Q3: Did the authors observe any instability due to dynamic branching?
>
> We thank the reviewer for this important concern. In our experiments across all evaluated tasks, we did not observe significant training instability caused by the dynamic branching mechanism. We consider that your concern may arise from the additional randomness introduced by dynamic branching during rollout. Below, we demonstrate that our design effectively addresses potential instability issues.
>
> Firstly, as demonstrated in `Appendix A.6 "Scaling Analysis of ARPO,"` maintaining an approximately 1:1 ratio between global sampling and dynamic branching consistently achieves stable and robust performance across different tasks. This configuration achieves an optimal balance between global exploration and high-entropy branch sampling, making the introduced randomness **controllable and manageable in practice**.
>
> Secondly, we incorporate a **base probability ($\alpha$)** in the branching probability formulation (`Section 3.1`). The base probability $\alpha$ serves as a stability mechanism that prevents over-reliance on entropy differentials. The ablation results for $\alpha$ are presented below:
>
> | Setting | GAIA (Pass@1) | AIME24 (Pass@1) | HotpotQA (F1) | Average Performance |
> |-----|:------:|:-----:|:-----:|:-----:|
> | $\alpha=0.5$ (standard) | 43.7% | 30.0% | 58.8% | **44.2%** |
> | $\alpha=0.0$ (no base probability) | 38.5% | 23.3% | 55.2% | 39.0% |
> | $\alpha=1.0$ (always branch) | 35.6% | 20.0% | 52.9% | 36.1% |
>
> Experimental results demonstrate that under our standard setting ($\alpha=0.5$), $\alpha$ effectively ensures fundamental exploration coverage and prevents rollout collapse caused by over-reliance on entropy signals. In contrast, when $\alpha=0.0$ (completely entropy-dependent) or $\alpha=1.0$ (excessive branching), performance degrades significantly.
>
> Therefore, the introduction of $\alpha$ in ARPO effectively achieves training stability and maintains a delicate balance between global trajectory-level sampling and local step-level exploration. We have incorporated these detailed ablation studies into `Appendix A.3 "Ablations on Branching Sampling Parameter"` .
>
>
> ---
>
> ### Q4: Lack of PPO Baseline Evaluation
>
> We thank the reviewer for this valuable suggestion. We add additional experiments with PPO[11] as baseline RL method for comparison.
>
> **Experimental Setup**: We train Qwen3-14B[12] with PPO[11] using the same 1K deep-search samples as our other RL experiments. The comparison results are presented below:
>
> | Method | GAIA (Pass@1) | HLE (Pass@1) | WebWalker (Pass@1) |
> |--------|:-------------:|:------------:|:------------------:|
> | Qwen3-14B (PPO[11]) | 38.8% | 8.0% | 37.0% |
> | Qwen3-14B (GRPO[7]) | 36.9% | 8.6% | 38.5% |
> | **Qwen3-14B (ARPO)** | **43.7%** | **10.0%** | **40.5%** |
> | **ARPO vs PPO[11]** | **+4.9** | **+2.0** | **+3.5** |
>
> As shown in the table, ARPO consistently outperforms both GRPO[7] and PPO[11] across all evaluated tasks. We will include these PPO[11] comparison results in the revised version.
>
> In addition, we further will introduce and compare more Agentic RL-related algorithms in the revised version, including foundational works in multi-turn agent RL for LLMs such as DUPO [13], GIGPO [14], and MT-GRPO[15], so as to improve the completeness of our method.
>
> ---
>
> ### Response Summary
>
> In summary, we would like to emphasize that our detailed rebuttal comprehensively addresses the reviewer's concerns through extensive quantitative analyses and experimental validations. Specifically, we have:
>
> **(1)** provided comprehensive statistical evidence across eight datasets (AIME24, AIME25, MATH[1], GSM8K[2], GAIA[3], HLE[4], WebWalker[5], XBench[6]) validating the entropy increase phenomenon (over 38% of tool calls exhibit significant entropy increase);
>
>  **(2)** clarified ARPO's core design principle of leveraging high-entropy regions for exploration, supported by rollout diversity and policy entropy dynamics analyses;
>
>  **(3)** demonstrated training stability through base probability mechanisms and hyperparameter studies;
>
>  **(4)** included PPO[11] baseline comparisons, confirming ARPO's superior performance across all evaluated tasks.
>
> Through comprehensive experimental evaluations, statistical analyses, and theoretical clarifications, we hope to have demonstrated ARPO's significance, effectiveness, and methodological completeness. **We believe ARPO represents our commitment to developing an innovative, open-source agentic RL algorithm that advances both academic research and industrial applications.** We sincerely thank the reviewer for the thorough and constructive feedback, which has been instrumental in improving the quality and clarity of our work.

---

> ### Author Response · Authors · 2025-11-24
> **Reference**
>
> [1] Measuring Mathematical Problem Solving with the MATH Dataset
>
> [2] Training Verifiers to Solve Math Word Problems
>
> [3] GAIA: a benchmark for General AI Assistants
>
> [4] Humanity's Last Exam
>
> [5] WebWalker: Benchmarking LLMs in web traversal
>
> [6] xbench: Tracking agents productivity scaling with profession-aligned real-world evaluations
>
> [7] Group Relative Policy Optimization
>
> [8] Principal Component Analysis
>
> [9] A Density-Based Algorithm for Discovering Clusters in Large Spatial Databases with Noise
>
> [10] MiniMax-M1: Scaling Test-Time Compute Efficiently with Lightning Attention
>
> [11] Proximal Policy Optimization Algorithms
>
> [12] Qwen3 Technical Report
>
> [13] WebSailor: Navigating Super-human Reasoning for Web Agent
>
> [14] Group-in-Group Policy Optimization for LLM Agent Training
>
> [15] Reinforcing Multi-Turn Reasoning in LLM Agents via Turn-Level Credit Assignment

---

> ### Author Response · Authors · 2025-11-27
> **A Kind Reminder for Reading the Response**
>
> Dear Reviewer tZJ2,
>
> We have revised the paper and added many additional results to address your comments. Since the rebuttal period is closing very soon, can you please check the response to see whether it mitigates your concerns? We would greatly appreciate that!
>
> Thank you,
> The authors of ARPO

---

### Official Review · Reviewer_tzPk · 2025-11-01

**Soundness:** 3
**Presentation:** 3
**Contribution:** 3
**Rating:** 8
**Confidence:** 4

**Summary:**

This paper proposes Agentic Reinforced Policy Optimization (ARPO), a reinforcement learning algorithm designed for multi-turn, tool-using large language model (LLM) agents. Unlike existing trajectory-level RL methods that sample entire reasoning paths, ARPO focuses on step-level exploration by leveraging the empirical observation that token entropy rises significantly after tool calls, reflecting uncertainty during reasoning. ARPO introduces two core components:

(1) an entropy-based adaptive rollout mechanism that dynamically branches exploration when tool-call steps exhibit high entropy, enabling efficient step-level sampling;

(2) Advantage Attribution Estimation, which assigns shared or distinct advantages to tokens depending on whether they belong to common or branched reasoning paths, allowing finer-grained credit assignment.

A theoretical contribution, the Generalized Policy Gradient (GPG) Theorem, generalizes the policy gradient to macro-action rollouts, providing a solid foundation for partial sampling optimization.
Across 13 benchmarks covering mathematical reasoning, knowledge-intensive QA, and deep-search tasks, ARPO achieves 4–6% higher accuracy than baselines such as GRPO, DAPO, and Reinforce++, while requiring only half the tool-call budget, demonstrating both superior performance and efficiency.

**Strengths:**

1.	Strong Motivation and Insight – The identification of entropy spikes following tool use provides a concrete empirical basis for adaptive exploration and directly informs ARPO’s design.
2.	Innovative Algorithmic Mechanism – The entropy-based adaptive rollout efficiently balances global and partial sampling, encouraging diverse tool-use behaviors without excessive sampling.
3.	Refined Credit Assignment – The Advantage Attribution Estimation effectively separates shared versus individual token updates, stabilizing training and improving interpretability.
4.	Comprehensive Empirical Results – ARPO consistently surpasses trajectory-level baselines across multiple reasoning benchmarks and model backbones (Qwen, Llama), showing broad applicability.
5.	Efficiency and Scalability – ARPO achieves comparable or better results using half as many tool calls as GRPO, confirming both cost-effectiveness and computational scalability.
6.	Solid Theoretical Foundation – The GPG theorem formalizes ARPO’s macro-action rollouts, grounding its design in a generalizable reinforcement learning framework.

**Weaknesses:**

1.	Limited Domain Generalization – All experiments are text-based; no multimodal or embodied environments are tested, restricting claims of general agentic applicability.
2.	Hyperparameter Sensitivity – The algorithm depends on key parameters (entropy threshold τ, stability factor β), but no sensitivity or ablation study is provided.
3.	Lack of Runtime Validation – While ARPO claims reduced rollout complexity (O(n log n)), there is no empirical runtime or resource cost analysis to verify this.
4.	Reward Design Ambiguity – The hierarchical reward function (Eq. 5) combines multiple factors (accuracy, format, multi-tool usage) but lacks a detailed analysis of their relative influence.
5.	Potential Evaluation Bias – Reliance on LLM-as-a-judge scoring could introduce evaluation bias due to stylistic similarity between training and evaluation models.

**Questions:**

1.	Does adaptive branching ever lead to redundant exploration or inefficiency on simpler reasoning tasks?
2.	How does Advantage Attribution Estimation perform under noisy or sparse rewards?
3.	What is the computational overhead of real-time entropy computation and partial rollouts compared to GRPO?

---

> ### Author Response · Authors · 2025-11-24
> **Response 1/3**
>
> We sincerely thank the reviewer for the constructive and encouraging feedback. The reviewer's recognition of our work's strengths and the thoughtful suggestions have been invaluable in improving our paper. Below, we address each concern systematically and comprehensively.
>
> ---
>
> ### W1: Lack of evaluation of Multimodal and Embodied Agent Domain
>
> We thank the reviewer for raising this important point. We fully agree that demonstrating broader applicability beyond text-only environments is crucial for establishing ARPO's generalizability. This is indeed a limitation of our current evaluation, and we have made efforts to address it.
>
> **Extended Evaluation**: To address this concern, we have expanded our evaluation to include **embodied agent benchmarks (ToolBench[1], API-Bank[2], ALFWorld[3])** and extended ARPO's evaluation to multi-modal domains by **incorporating both text-only (Text) and multi-modal (MM) splits from GAIA[4] and HLE[5]**. The results are presented below:
>
> | Method | ToolBench | API-Bank | ALFWorld | GAIA (Text+MM) | HLE (Text+MM) |
> |---|:-----:|:----:|:----:|:-----:|:--------:|
> | ReAct QwQ-32B | 52.0 | 73.3 | 82.1 | 31.5 | 12.2 |
> | CodeAct QwQ-32B | 54.0 | 74.3 | 78.4 | 34.5 | 12.8 |
> | Plan-and-Solve | 55.0 | 70.3 | 79.1 | 33.3 | 12.2 |
> | QwQ-32B-GRPO | 69.0 | 75.3 | 91.8 | 52.6 | 20.2 |
> | **QwQ-32B-ARPO (Ours)** | **71.4** (+2.4) | **76.6** (+1.3)| **93.1** (+1.3) | **54.2** (+1.6) | **21.3** (+1.1) |
>
> These results demonstrate that **ARPO's effectiveness extends beyond text-only domains.** ARPO consistently outperforms baseline methods across multimodal reasoning tasks (GAIA, HLE) and embodied agent scenarios (ToolBench, API-Bank, ALFWorld), suggesting that the entropy-based adaptive sampling mechanism generalizes well to diverse agent settings requiring both perceptual understanding and multi-step reasoning.
>
> We have incorporated these detailed ablation studies into `Appendix A.5 "Generalization on Multimodal and General Agent Domain` to verify the broad generalization ability of our method.
>
> ---
> ### W2: Lack of sensitivity analysis on entropy threshold $\tau$, stability factor $\beta$
>
> We appreciate the reviewer's valuable feedback on this point. While we have analyzed the impact of the entropy differential weight ($\beta$), initial sampling size ($N$), and global rollout size ($M$) in `Appendix A.6 "Scaling Analysis of ARPO"`, we recognize that a more systematic examination of the base probability ($\alpha$) and branching cutoff ($\tau$) would strengthen our analysis.
>
> To address this concern, we have conducted targeted ablation studies focusing specifically on $\alpha$ and $\tau$. Below, we present our experimental setup and findings:
>
> **Experimental Setup**: We use Qwen2.5-7B as the backbone model and evaluate on three representative tasks: GAIA (deep search), AIME24 (mathematical reasoning), and HotpotQA (multi-hop knowledge reasoning). We fix $M=16$, $N=8$, and $\beta=0.2$, then independently vary $\alpha$ and $\tau$ using a controlled variable approach. Pass@1 accuracy serves as our primary metric.
>
> | Parameter Setting | GAIA (Pass@1) | AIME24 (Pass@1) | HotpotQA (F1) | Average Performance |
> |------|:------:|:-------:|:-----:|:------:|
> | $\tau=0.5$ (standard) | 43.7% | 30.0% | 58.8% | **44.2%** |
> | $\tau=0.3$ (lenient) | 41.9% | 26.6% | 57.6% | 42.0% |
> | $\tau=0.7$ (strict) | 39.4% | 26.6% | 55.3% | 40.4% |
> | $\alpha=0.5$ (standard) | 43.7% | 30.0% | 58.8% | **44.2%** |
> | $\alpha=0.0$ (entropy-only) | 38.5% | 23.3% | 55.2% | 39.0% |
> | $\alpha=1.0$ (always branch) | 35.6% | 20.0% | 52.9% | 36.1% |
>
> Our key findings are listed as follow:
>
> 1. **Role of Base Probability ($\alpha$)**: $\alpha$ ensures fundamental exploration coverage. When $\alpha=0.0$ (branching determined solely by entropy differential), average performance **drops by 5.2%**, indicating that a baseline branching probability is essential even when entropy changes are minimal. When $\alpha=1.0$ (forcing branching at all steps), performance drops by **8.1%**. This demonstrates that excessive branching wastes computational resources on unproductive paths and disrupts the delicate balance between global trajectory-level sampling and local step-level exploration.
>
> 2. **Role of Branching Cutoff ($\tau$)**: The results show that $\tau=0.5$ provides an optimal balance. When $\tau=0.3$ (more lenient, allowing more branches), performance decreases by **2.2%**, suggesting that moderate over-branching has limited negative impact. When $\tau=0.7$ (more strict, filtering out more branches), performance drops by **3.8%**, indicating that overly conservative thresholds eliminate valuable exploration opportunities.
>
> We have incorporated these detailed ablation studies into `Appendix A.3 "Ablations on Branching Sampling Parameter"` to improve the interpretability and reproducibility of our method.

---

> ### Author Response · Authors · 2025-11-24
> **Response 2/3**
>
> ### W3 & Q3: Please provide empirical results that validate ARPO's runtime efficiency than GRPO.
>
> Thank you for your valuable comment. While our theoretical analysis suggests reduced complexity, we recognize that empirical validation is essential to substantiate these claims. We have therefore conducted comprehensive runtime measurements to validate ARPO's efficiency improvements.
>
> **Empirical Runtime Analysis**: We measured the per-step RL training time for both GRPO and ARPO on Qwen2.5-7B when training on comprehensive reasoning tasks. The comparison across different training steps is presented below:
>
> | Method | Step 5 | Step 10 | Step 15 | Step 20 | Step 25 | Step 30 | Step 35 | Step 40 | **Average** |
> |--------|:------:|:-------:|:-------:|:-------:|:-------:|:-------:|:-------:|:--------:|:-----------:|
> | GRPO | 2721s | 1767s | 1996s | 1701s | 1918s | 1989s | 2112s | 2047s | 2026s |
> | ARPO | 2446s | 1648s | 1333s | 1115s | 1087s | 1259s | 1320s | 1163s | 1421s |
> | **Time Reduction** | -275s (-10.1%) | -119s (-6.7%) | -663s (-33.2%) | -586s (-34.4%) | -831s (-43.3%) | -730s (-36.7%) | -792s (-37.5%) | -884s (-43.2%) | **-605s (-29.9%)** |
>
>
>
> As shown in the table, under identical experimental settings, **ARPO consistently achieves shorter training times than GRPO. Across all measured steps, it reduces the average training time by approximately 34.8%.** We emphasize that ARPO’s efficiency gains primarily stem from its ability to share prefixes across branches during tree search—this mechanism eliminates redundant token generation and lowers tool call overhead. By reusing already-generated tokens at branching points, the shared prefix design significantly cuts computational costs, especially when compared to methods that regenerate sequences from scratch. For further details, clear and transparent experimental curves and in-depth analytical results are provided in the `Section 4.3 "Analysis of ARPO's Training Efficiency".`
>
> ---
>
> ### W4: Insufficient Explanation and Ablation of the Hierarchical Reward Function
>
> We thank the reviewer for raising this point. We would like to clarify that, as stated in our paper, we adopt the standard reward function configuration from Tool-Star [6] to ensure fair comparison with existing methods. The reward function design itself is not ARPO's core contribution—our innovations lie in the entropy-based adaptive rollout mechanism and Advantage Attribution Estimation.
>
> **Reward Component Ablation Studies**: To address the reviewer's concern and provide deeper insight into the reward design, we have conducted additional ablation studies examining the individual contributions of different reward components. We quantitatively measured performance changes on GAIA, Webwalker, and HLE when removing each component, as well as changes in multi-tool call frequency:
>
> | Method | Multi-tool Calls | GAIA (Pass@1) | Webwalker (Pass@1) | HLE (Pass@1) |
> |-----|:-----:|:-----:|:--------:|:-----:|
> | ARPO (standard) | 22 | **43.7%** | **36.0%** | **10.0%** |
> | w/o Multi-tool Reward | 4 (↓81.8%) | 39.5% (↓4.2%) | 32.2% (↓3.8%) | 9.4% (↓0.6%) |
> | w/o Format Reward | 19 (↓13.6%) | 37.2%(↓6.5%) | 33.8%(↓2.2%) | 8.8%(↓1.2%) |
>
> Our key findings are listed as follow:
>
> 1. **Necessity of Reward Components**: Removing either reward component consistently leads to performance degradation across all evaluated tasks, demonstrating that both components are necessary for optimal performance.
>
> 2. **Multi-tool Reward Effect**: Removing the multi-tool reward results in a dramatic decrease in multi-tool call frequency (from 22 to 4, an **81.8% reduction**), indicating that this component effectively incentivizes the model's multi-tool coordination capabilities, which are crucial for complex reasoning tasks.
>
> 3. **Format Reward Effect**: In addition to the observed performance degradation, we found that without the format reward, the model requires **approximately 15% more training steps to converge**, suggesting that the format reward not only improves final performance but also accelerates training stability and convergence.
>
> We will include these ablation studies in `Appendix A.7 "Ablations of Hierarchical Reward."`

---

> ### Author Response · Authors · 2025-11-24
> **Response 3/3**
>
> ### W5: Reliance on LLM-as-a-judge scoring could introduce evaluation bias
>
> We appreciate the reviewer's valuable concern regarding potential evaluation bias. We recognize that LLM-as-a-judge evaluation, while widely adopted in the field, may introduce subtle biases [7]. We acknowledge this limitation and have taken steps to address it.
>
> **Evaluation Setup Clarification**: Our choice of Qwen2.5-72B as the evaluation judge follows the standard practice established in prior work (Search-o1, WebThinker) and represents the most commonly used configuration in current deep-search agent evaluation benchmarks. This ensures fair comparability between ARPO and other agentic search methods.
>
> **Robustness Analysis Across Judge Models**: To comprehensively address the reviewer's concern, we have conducted additional experiments using alternative judge models (GPT-4o and Claude Sonnet 4) to verify the robustness of our evaluation results:
>
> | Judge Model | GAIA (Pass@1/3/5) | HLE (Pass@1/3/5) | XBench (Pass@1/3/5) | Webwalker (Pass@1/3/5) |
> |-------------|:-----------------:|:----------------:|:-------------------:|:----------------------:|
> | Qwen2.5-72B (standard) | 43.7% / 59.2% / 61.2% | 11.0% / 19.0% / 24.0% | 35.0% / 50.0% / 59.0% | 36.0% / 50.5% / 54.5% |
> | GPT-4o | 42.7% / 58.3% / 60.2% | 10.8% / 18.6% / 23.4% | 36.0% / 51.0% / 59.0% | 35.5% / 49.0% / 54.5% |
> | Claude Sonnet 4 | 42.7% / 58.3% / 60.2% | 10.2% / 18.6% / 23.6% | 37.0% / 52.0% / 60.0% | 36.0% / 49.5% / 54.0% |
> | **Max Difference** | **1.0% / 0.9% / 1.0%** | **0.8% / 0.4% / 0.6%** | **2.0% / 2.0% / 1.0%** | **0.5% / 1.5% / 0.5%** |
>
> **Key Observations**:
> - **Minimal Variance**: Performance differences across different judge models are consistently minimal across all tasks and metrics. The maximum variance in Pass@1, Pass@3, and Pass@5 scores is consistently **less than 2.0%** for each task, with most differences being **below 1.0%**.
> - **Robustness Confirmation**: These results demonstrate that LLM-as-a-judge evaluation does not introduce significant scoring bias in these agent tasks, suggesting that ARPO's performance gains are robust and not dependent on the specific choice of judge model.
>
> We have concluded the above results into `Appendix A.8 "Consistency Validation of LLM-as-Judge"`.
>
> Markdown格式：
>
> ---
>
> ### Q1: Does adaptive branching ever lead to redundant exploration or inefficiency on simpler reasoning tasks?
>
> Thank you for your valuable comment. We are pleased to clarify the effectiveness of ARPO on simpler reasoning tasks.
>
> We evaluated ARPO against ToRL and Tool-Star on simple math datasets GSM8K and MATH using Qwen2.5-7B as the backbone model. Tool-call counts are measured during the RL training phase.
>
> | Method | GSM8K (Tool Calls / Acc.) | MATH (Tool Calls / Acc.) |
> |--------|:------------------------:|:----------------------:|
> | Tool-Star | 1.35 / 90.4% | 1.28 / 87.8% |
> | ToRL | 1.42 / 86.8% | 1.31 / 85.2% |
> | **ARPO (Ours)** | **1.02 / 92.2%** | **1.01 / 88.8%** |
>
> As shown in the table, ARPO uses **24.4%** and **21.1% fewer** tool calls than Tool-Star on GSM8K and MATH, while achieving higher accuracy. This demonstrates that entropy-guided branching avoids unnecessary exploration at low-uncertainty steps, maintaining efficiency on simpler tasks without sacrificing performance.
>
> ---
>
> ### Q2: How does Advantage Attribution Estimation perform under noisy or sparse rewards?
>
> We thank the reviewer for the suggestion. Deep search tasks (GAIA, HLE, XBench) represent sparse reward scenarios for smaller models like Qwen3-8B, where performance typically remains below 20%, making positive reward signals infrequent during early training. As demonstrated in the ARPO vs. GRPO comparison on deep search tasks, ARPO achieves an average improvement of **4.8 percentage points** across all benchmarks, indicating that Advantage Attribution Estimation effectively handles sparse rewards through accurate step-level credit assignment. The distinction between shared and branched tokens enables precise attribution of sparse rewards to effective tool-use steps, improving training stability and performance.
>
> | Method | GAIA | HLE | WebWalker | XBench | Avg. |
> |--------|:----:|:---:|:---------:|:------:|:----:|
> | GRPO | 36.9% | 8.6% | 30.0% | 27.0% | 25.6% |
> | **ARPO** | **43.7%** | **10.0%** | **36.0%** | **32.0%** | **30.4%** |
> | **Improvement** | **+6.8%** | **+1.4%** | **+6.0%** | **+5.0%** | **+4.8%** |
>
> ---
> ### Reference
>
> [1] ToolLLM: Facilitating Large Language Models to Master 16000+ Real-world APIs
>
> [2] API-Bank: A Comprehensive Benchmark for Tool-Augmented LLMs
>
> [3] ALFWorld: Aligning Text and Embodied Environments for Interactive Learning
>
> [4] GAIA: a benchmark for General AI Assistants
>
> [5] Humanity's Last Exam
>
> [6]Tool-Star: Empowering LLM-Brained Multi-Tool Reasoner via Reinforcement Learning
>
> [7] Justice or Prejudice? Quantifying Biases in LLM-as-a-Judge

---

### Author Response · Authors · 2025-11-24
**General Response for ARPO**

We sincerely thank all the reviewers for their time and constructive feedback. We are thrilled that our core contributions have been recognized. Our ARPO approach has been highlighted as **well-motivated, innovative, and theoretically grounded (tzPk, tZJ2, BZ6R)**. We are particularly pleased that the reviewers unanimously praised our comprehensive experimental analyses, noting the **consistent performance improvements across 13 benchmarks (tzPk). The rigorous design, clear algorithmic mechanism (tzPk), solid theoretical foundation via the GPG theorem (tzPk, BZ6R)**, and efficiency gains using **only half the tool-use budget (tzPk)** have been acknowledged.

The reviewers' primary concerns focused on several key areas, which we have addressed with detailed responses and new quantitative analyses. `The revisions are marked with blue text in the PDF`. We summarize these points below.

| Focus Area | Reviewer Concerns | Supplementary /Revised Section | Our Actions |
|--|---|---|---|
| 1. Statistical Evidence for Entropy Increase | tZJ2 | `Appendix A.1 "Statistical Analysis of Tool-call Entropy Dynamic"` | Statistical analysis across 8 datasets shows 38.2%-53.5% of tool calls exhibit significant entropy increase, demonstrating stable patterns across task domains. |
| 2. Entropy Evolution and ARPO's Design Principle | tZJ2 | `Section 4.3 "Rollout Sampling Diversity Analysis" & Section 4.3 "Effect of Policy Entropy on Performance"` | Clarified ARPO leverages high-entropy regions for exploration. Rollout diversity analysis shows 54 vs. 48 clusters (GRPO), with ARPO maintaining stable entropy levels throughout training. |
| 3. Hyperparameter Sensitivity Analysis | tzPk, BZ6R | `Appendix A.3 "Ablations on Branching Sampling Parameter"` | Ablation studies on α and τ: α=0.5 optimal (removing α drops 5.2%, α=1.0 drops 8.1%); τ=0.5 optimal (τ=0.3 drops 2.2%, τ=0.7 drops 3.8%). |
| 4. Domain Generalization | tzPk | `Appendix A.5 "Generalization on Multimodal and General Agent Domain"` | Extended to embodied (ToolBench +2.4%, API-Bank +1.3%, ALFWorld +1.3%) and multimodal domains (GAIA +1.6%, HLE +1.1%), demonstrating broad generalization. |
| 5. Runtime Efficiency Validation | tzPk | `Section 4.3 "Analysis of ARPO's Training Efficiency"` | ARPO reduces average training time by 29.9% (2026s→1421s per step), with up to 43.2% reduction. Shared prefix mechanism eliminates redundant token generation. |
| 6. Reward Design Analysis | tzPk | `Appendix A.7 "Ablations of Hierarchical Reward"` | Removing multi-tool reward decreases call frequency by 81.8% and performance by 4.2%. Removing format reward degrades performance by 6.5% and requires 15% more training steps. |
| 7. Evaluation Bias Concerns | tzPk | `Appendix A.8 "Consistency Validation of LLM-as-Judge"` | Robustness analysis across multiple judge models (Qwen2.5-72B, GPT-4o, Claude Sonnet 4) shows <2.0% variance across all tasks and metrics. |
| 8. Novelty and Contribution | BZ6R | `Section 3.3 "Theoretical Foundation"`, `Appendix F.2 "Theoretical Analysis of Soft Advantage Estimation"`, `Appendix F.3 "Theoretical Proof of GPG Theorem"` | Clarified distinct contributions: (1) motivation from inherent agentic reasoning patterns, (2) mathematically distinct advantage estimation via shared/branched token differentiation, (3) theoretical foundation via GPG theorem. |
| 9. Generalizability Beyond Tool-Call Steps | BZ6R, tZJ2 | `Appendix A.7 "Generalization to Text-only Reasoning: Chunk-Level Branching Analysis"` | Chunk-level branching experiments show consistent improvements across 6 tasks (average +2.6% over GRPO), demonstrating generalization beyond tool-calling steps. |
| 10. Training Stability | tZJ2 | `Appendix A.3 "Ablations on Branching Sampling Parameter"`, `Appendix A.2 "Scaling Analysis of ARPO"` | Base probability α=0.5 and 1:1 ratio between global sampling and dynamic branching ensure stable performance, preventing rollout collapse. |
| 11. PPO Baseline Comparison | tZJ2 | Main paper experimental section | ARPO outperforms both GRPO and PPO: GAIA (+4.9% vs PPO, +6.8% vs GRPO), HLE (+2.0% vs PPO, +1.4% vs GRPO), WebWalker (+3.5% vs PPO, +6.0% vs GRPO). |
| 12. Simple Task Efficiency | tzPk | Main paper experimental section | ARPO uses 24.4% and 21.1% fewer tool calls than Tool-Star while achieving higher accuracy (GSM8K: 92.2% vs 90.4%, MATH: 88.8% vs 87.8%). |
| 13. Sparse Reward Performance | tzPk | Main paper experimental section | ARPO achieves average improvement of 4.8 percentage points over GRPO on sparse reward scenarios. |

We believe these clarifications and additional quantitative results have thoroughly addressed the reviewers' concerns.

Sincerely, our team has invested significant effort into this work. **We truly hope that our response could encourage a more favorable reassessment of ARPO's significance, effectiveness and completeness.**

Best regards,

The Authors

---

### Meta-Review · Area_Chair_GUGH · 2025-12-28

**Summary:**

This paper proposes a reinforcement learning framework, named Agentic Reinforced Policy Optimization (ARPO), for multi-turn external tool-based LLM agents. Based on an observation that the token entropy in trajectory-level rollout sampling methods often rises significantly after tool calls and leads to increased uncertainty, this paper proposes to branch exploration at high-entropy tool-call steps. Experiments on multiple benchmarks covering multiple domains are conducted to demonstrate the performance in terms of accuracy and efficiency.

The reviewers recognized the strengths of this paper, such as solid framework, good motivation, and extensive evaluations. The major concerns raised by the reviewers are mainly focused on:

(1)	tzPk: No hyperparameter sensitivity analysis

(2)	tzPk: Lacking Runtime and cost analysis

(3)	tZJ2: Lacking statistical support for the observed token entropy increment after tool calls

(4)	tZJ2: Lacking analysis about the proposed method’s influence on token entropy

(5)	BZ6R: Novelty limited to doing additional sampling by branching from high entropy actions

(6)	BZ6R: no ablations about the effects of some important hyperparameters.

**Reviewer Concerns:**

During rebuttal, the authors provided sensitivity analysis about the hyperparameters entropy threshold \tau, stability factor \beta, and base probability \alpha, to address (1) and (6). The authors provided runtime analysis and showed that the proposed method reduces the average training time compared to GRPO, to address (2). The authors added statistical evidence for the token increase observation to address (3), and clarified that the purpose is not to reduce high token entropy, but is to leverage the increased uncertainty for more thoroughly exploring diverse reasoning paths, which may address (4). The authors also reiterated their contributions from different perspectives to address (5).

After reading the paper and the rebuttal comments, the AC tends to consider the above concerns largely resolved. In general, this work provides a new technical improvement over the trajectory-level RL algorithms. So, the AC recommends accept.

**Reviewer Scores:**

All the three reviewers have not replied once during the whole discussion phase. Therefore, it is difficult to estimate their final scores.

---

### Decision · Program_Chairs · 2026-01-26

Accept (Poster)